# *Cis* inhibition of NOTCH1 through JAGGED1 sustains embryonic hematopoietic stem cell fate

Roshana Thambyrajah [1,2,3] ✉, Maria Maqueda [1], Wen Hao Neo [4], Kathleen Imbach[2], Yolanda Guillén[1], Daniela Grases[2], Zaki Fadlullah[4], Stefano Gambera [5], Francesca Matteini [6,7], Xiaonan Wang [8,11], Fernando J. Calero-Nieto [8], Manel Esteller [2,3,9,10], Maria Carolina Florian [3,6,7], Eduard Porta[2], Rui Benedito [5], Berthold Göttgens [8], Georges Lacaud[4], Lluis Espinosa [1,3] & Anna Bigas [1,2,3] ✉

Hematopoietic stem cells (HSCs) develop from the hemogenic endothelium (HE) in the aorta- gonads-and mesonephros (AGM) region and reside within Intra-aortic hematopoietic clusters (IAHC) along with hematopoietic pro- genitors (HPC). The signalling mechanisms that distinguish HSCs from HPCs are unknown. Notch signaling is essential for arterial specification, IAHC for- mation and HSC activity, but current studies on how Notch segregates these different fates are inconsistent. We now demonstrate that Notch activity is highest in a subset of, GFI1 + , HSC-primed HE cells, and is gradually lost with HSC maturation. We uncover that the HSC phenotype is maintained due to increasing levels of NOTCH1 and JAG1 interactions on the surface of the same cell (*cis*) that renders the NOTCH1 receptor from being activated. Forced activation of the NOTCH1 receptor in IAHC activates a hematopoietic differ- entiation program. Our results indicate that NOTCH1-JAG1 *cis*-inhibition pre- serves the HSC phenotype in the hematopoietic clusters of the embryonic aorta.

Hematopoietic stem cells (HSCs) have the exceptional ability to self-renew and re-establish the entire blood system after injury or transplantation, making them very attractive to treat blood dis-orders. The initial HSCs are detected during mid-gestation (E10-E12) in the trunk of the embryo, where the aorta, gonads and mesone-phros (AGM) co-localize. At the site of their origin, they are detected in intra-aortic hematopoietic clusters (IAHC), clusters of hemato-poietic cells that reach into the lumen of the dorsal aorta (DA)[1–4]. Although several hundreds of such hematopoietic cells (HSPC) are organized in 30-40 IAHC, only a small minority of these cells show HSC activity in transplantation assays[5–7]. Markers that distinguish these early HSCs from multipotent progenitors (hereafter referred to

[1]Program in Cancer Research. Institut Hospital del Mar d'Investigacions Mèdiques, CIBERONC, Barcelona, Spain. [2]Josep Carreras Leukemia Research Institute, Barcelona, Spain. [3]Centro de Investigacion Biomedica en Red (CIBER), Madrid, Spain. [4]Cancer Research UK Stem Cell Biology Group, Cancer Research UK Manchester Institute, The University of Manchester, Manchester, UK. [5]Molecular Genetics of Angiogenesis Group. Centro Nacional de Investigaciones Cardiovasculares (CNIC), Madrid, Spain. [6]Stem Cell Aging Group, Regenerative Medicine Program, The Bellvitge Institute for Bio-medical Research (IDIBELL), L'Hospitalet de Llobregat, Barcelona, Spain. [7]Program for advancing the Clinical Translation of Regenerative Medicine of Catalonia (P-CMR[C]), Barcelona, Spain. [8]Department of Haematology, Wellcome - MRC Cambridge Stem Cell Institute, Cambridge Biomedical Campus, Cambridge, UK. [9]Institució Catalana de Recerca i Estudis Avançats (ICREA), Barcelona, Catalonia, Spain. [10]Physiological Sciences Department, School of Medicine and Health Sciences, University of Barcelona (UB), Barcelona, Catalonia, Spain. [11]Present address: School of Public Health, Shanghai Jiao Tong University, School of Medicine, Shanghai, China. ✉e-mail: roshanathambyrajah@gmail.com; abigas@researchmar.net

as HPC) are only starting to emerge[8]. Indeed, very recent studies demonstrate that the hematopoietic system with its established hierarchy, i.e., with long/short term (LT/ST-) HSCs and HPCs are already present in the AGM[8–10]. The recent findings suggests that all these hematopoietic cells migrate to the fetal liver to multiply simultaneously, as opposed to the (LT)-HSCs pool derived from the AGM generating the entire blood system in the fetal liver[9,10]. This updated model implies that certain molecular mechanisms must be in place to support the emergence of these diverse blood populations within a short length of time in the AGM.

All IAHC are derived from hemogenic endothelial (HE) cells, a specialised endothelium that is embedded with the DA, but already expresses essential transcription factors of HSPC, including *Gata2*, *Runx1* and *Gfi1*[11–16]. These HE cells undergo a transformation to a hematopoietic phenotype, termed endothelial to hematopoietic transition (EHT) whereby they round up and bud into the lumen with concomitant proliferation and thereby decrease endothelial marker gene expression[1,17–21]. During this process, the cells undergoing EHT start to gain the hematopoietic markers *cKIT* and CD41 followed by CD45 in a small minority of IAHC, which categories HSCs in the AGM [22–24].

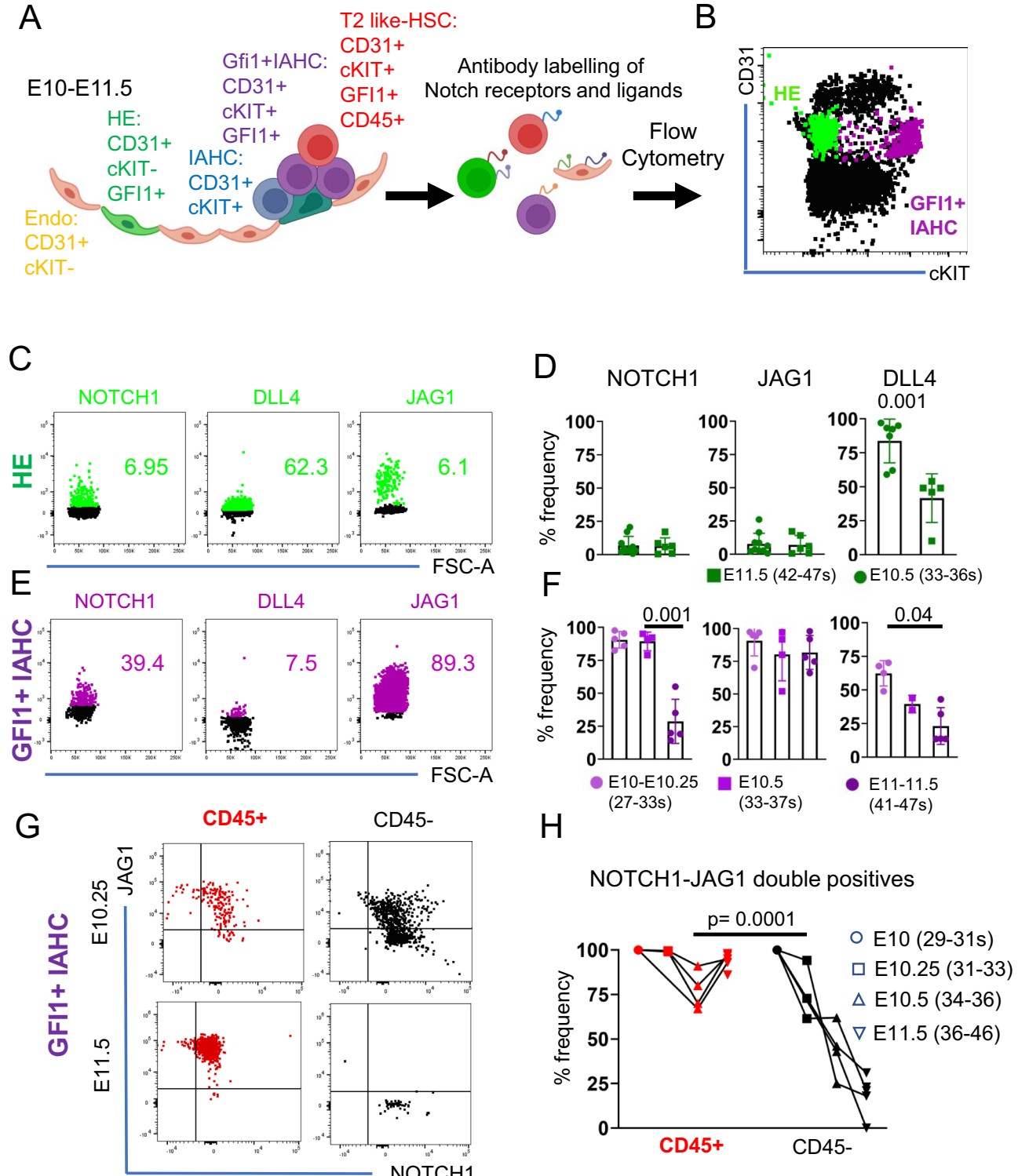

**Fig. 1 | NOTCH1 and JAG1 are co-expressed in T2-HSCs. A** scheme of the IAHC formation in the mouse embryo. GFI1 + HE cells undergo EHT and form IAHC within the dorsal aorta. Within these clusters, the GFI1 + IAHC fraction contains all HSC activity. The HSC population can be further restricted by including CD45 as a marker of T2-HSCs (T2-like HSC). **B** Representative Flow cytometry plot of AGM lysates stained for CD31 and cKIT. The gate for GFI1+ cells within each population is superimposed onto the plot. **C** Representative Flow Cytometry plots of HE with superimposed gates for indicated notch signaling molecule. **D** Quantification of NOTCH1, JAG1 and DLL4 positive cells within the HE population at E10.5 and E11.5. 2-5 AGMs were pooled for each data point in 4 independent experiments (E10.5 $n = 18$ embryos, E11.5 $n = 22$ embryos). Statistical significance was calculated with two-tailed t-tests. Vertical error bars indicate the mean and standard deviation values. Source data are provided as a source data file. **E** Exemplary Flow Cytometry plots of GFI1 + IAHC with superimposed gates for indicated notch signaling molecule. **F** Quantification of NOTCH1, JAG1 and DLL4 positive cells within the GFI1 +

IAHC population at E10, E10.5 and E11.5. 2-18 AGMs were pooled for each data point in 4 independent experiments (E10 (27-33 s) $n = 32$ embryos, E10.5 $n = 18$ embryos, E11.5 $n = 22$ embryos). Statistical significance was calculated with two-tailed t-tests. Vertical error bars indicate the mean and standard deviation values. Source data are provided as a source data file. **G** Representative Flow Cytometry plots showing JAG1 and NOTCH1 levels within GFI1 + IAHC (CD31+cKIT+GFI1 + ) sub gated for CD45 at E10.25 and E11.5. NOTCH1-JAG1 double positive cells in the CD45+ fraction is highlighted in red. **H** Line chart of Notch1/JAG1 double positive cells within CD45 + /- IAHC (CD31+cKIT + ) between E10 and E11.5. 2-5 AGMs were pooled for each data point in E10 (27-28 s, $n = 8$ embryos, E10.25 (31-33 s, $n = 24$ embryos, E10.5 (34-36 s, $n = 18$ embryos and E11.5 (36-46, $n = 22$ embryos) 4 independent experiments for each time point. Statistical significance was calculated with two-tailed t-tests. Vertical error bars indicate the mean and standard deviation values. Source data are provided as a source data file. (s: number of somites).

Moreover, more stringent markers, including SCA1 and EPCR can be added to enrich the IAHC for HSCs (T2-HSCs)[25,26]. Numerous recent publications have analysed the stepwise progression of HE cells to HSPC at different developmental stages by single cell RNA sequencing[16,18,25,27,28]. Remarkably, many of them identified the expression of Notch signalling molecules and Notch targets as characteristic for arterial endothelium, HE and IAHC, indicating an essential role in these cell populations[18,28].

The Notch signalling pathway is highly conserved in metazoan and controls cell fate decisions during embryonic development that is initiated by cell-cell contact[29]. In mammals, there are four Notch receptors (*Notch1–4*) and five ligands: three Delta ligands (*Dll1*, *Dll3*, and *Dll4*) and two Jagged ligands (*Jag1* and *Jag2*)[30]. Typically, a cell displays one of the Notch receptors and its neighboring cell interacts through a Notch ligand which mediates Notch activation in the receptor bearing cell, thereby releasing the Notch Intracellular Domain (NICD), which then translocate into the nucleus to activate the transcriptional repressor genes, *Hes/Hey* in vertebrates which in turn can repress genes driving cell specification, cell differentiation, and cell cycle arrest[31].

Most studies have described Notch receptor and ligands interactions between neighbouring cells (*trans* interactions) that ultimately leads to cell fate segregation within a population by lateral induction or inhibition[32,33]. However, emerging studies postulate and demonstrate that receptor and ligand can be co-expressed by the same cell (in *cis*)[34–37]. This *cis* conformation has two advantages for the cell. Firstly, it can switch between different cell fates independent of *trans* activation through expressing ligands in *cis* that can be activating or inhibiting, and secondly, it can shield, reduce or prevent *trans*-activation by sequestering free receptors. Adding to the complexity of Notch signalling, the affinity of the Notch receptors to the ligands can by either weakened or enhanced by glycosylation of the extracellular part of the receptor by Fringe proteins (MANIC, LUNATIC and RADICAL)[38]. Whilst both LUNATIC and MANIC enhance binding of DLL1 to NOTCH1, RADICAL (*Rfng*) also enhances binding to JAG1[39] and, facilitates *cis* activity of NOTCH1 for JAG1[34].

Since Notch signalling is crucial for both arterial specification and IAHC (including HSC activity), it has been challenging to uncouple Notch signaling requirements for arterial identity from hematopoietic commitment[40]. In embryonic *Notch1*-chimeras, *Notch1*-deficient cells fail to contribute to haematopoiesis after E15.5, suggesting that *Notch1* is needed in hematopoietic cell intrinsically for their specification[41,42]. Counterintuitively, transgenic Notch activity tracing mouse models defined lower Notch activity in IAHC compared to their arterial surrounding[43] and blocking DLL4 with a specific antibody at the time of HSC emergence increases HSC frequency[17]. However, both complete and endothelial specific KO of the notch ligand *Jag1* show a specific loss of IAHC and HSC activity, although the artery formation is intact[44,45]. Finally, maturing T2-HSCs are Notch independent[46]. It is therefore

unclear how these seemingly contradictive phenotypes can be consolidated. Specifically, it is not known how lower Notch activity during HSC maturation through *Notch1* is established.

Here, we studied the dynamic protein levels of notch receptors and ligands by FACS during HE to IAHC transition. We find profound differences in the distribution of the ligands, DLL4 and JAG1. DLL4 is predominantly present at early stages and more specifically in HE, whereas JAG1 is detected robustly in IAHC and its levels are sustained from E10.25 to E11.5. By capturing the surface expression of Notch signaling molecules, NOTCH1, DLL4 and JAG1 at the time of FACS assisted purification of cells (Index FACS) we were able to link their protein expression profile to hematopoietic fate by single cell RNA seq. Remarkably, we identified a subset of GFI1 + HE that expresses high levels of Notch target genes and suggesting high Notch activity. This small population of Notch activated HE cells proceed to cluster towards T2-HSCs and already express markers of HSC fate. We visualized the interaction of NOTCH1 and DLL4/JAG1 by Proximity ligation assay. We found NOTCH1 and JAG1 interactions accumulated as foci in IAHC, and these interactions could be further identified as *cis* conformation. Furthermore, our single cell RNA seq data set indicated the expression of *Rfng* in a subset of T2-HSCs. By Immunohistochemistry (IHC) and FACS analysis we show that RFNG is specific for the T2-HSC population. RFNG knockdown AGMs have reduced numbers of HSCs and the NOTCH1-JAG1 *cis* interactions are reduced, suggesting that RFNG favors NOTCH1-JAG1 in *cis* and that this conformation is essential for HSC maintenance. Finally, we deregulated this NOTCH1-JAG1 cis interaction by culturing the cells ex vivo in the presence of recombinant JAG1 protein (Fc-JAG1). Using a nascent RNA capture assay, we found that genes associated with lineage differentiation and cell cycle are the main networks regulated by NOTCH1-JAG1 in *cis* conformation in GFI1 + IAHC.

Altogether, we uncovered an aspect of the role of Notch signalling in HSC biology where RFNG in HSCs facilitates NOTCH1-JAG1 interactions in *cis* that fine tunes the differentiation and cell cycle kinetics to a stem cell state.

## Results

### Notch receptors and ligand proteins are dynamically expressed in the AGM subpopulations

Recent availability of antibodies against different members of the Notch family allows the precise characterization of protein levels in rare populations at a single cell level. We characterized the presence of Notch receptors and ligands in the AGM subpopulations during hematopoietic development (E10.25-E11.5). We detected the presence of most of the ligands and receptors at E10.5, and a decrease at E11.5 was observed for NOTCH1 and DLL4 in the endothelium (CD31+cKIT-CD45-) and IAHC (CD31+cKIT + ), as well as for JAG1 in CD45 + IAHC (Supplementary Fig. 1 and Supplementary Data 1). Based on the literature and our previous work[42–44,46], we focused our subsequent analysis on NOTCH1, NOTCH2 receptors and JAG1 and DLL4 ligands.

In order to identify the HE and to restrict the HSC-containing IAHC, we used the *Gfi1:tomato* mouse line (Fig. 1A, B). The GFI1 reporter marks a sub-population of the CD31+cKIT- fraction that consists of HE, and a subpopulation of IAHC cells that contain all HSC activity[11] (Fig. 1B). On average, a large majority of cells in the HE subpopulation (80%) only express DLL4 on their surface, and about 15-20% are NOTCH1+ cells (Fig. 1C, D; Supplementary Fig. 3A). Expression of DLL4

in HE and GFI1 + IAHC steadily decreases from E10.5 to E11.5 (Fig. 1D, Supplementary Figs. 2 and 3A), and NOTCH1 expression in GFI1 + IAHC suddenly decreases at late AGM stages (Fig. 1E, F, Supplementary Figs. 2 and 3A). In stark contrast, JAG1 protein levels are low in HE (Fig. 1 C, D, Supplementary Figs. 1 and 2A, B), but is the most abundant ligand in GFI1 + IAHC (Fig. 1E, F, Supplementary Figs. 1 and 2A, B). In fact, T2-HSCs, defined as CD31+cKIT+GFI1 + CD45+ (Fig. 1 A) (ref. 25),

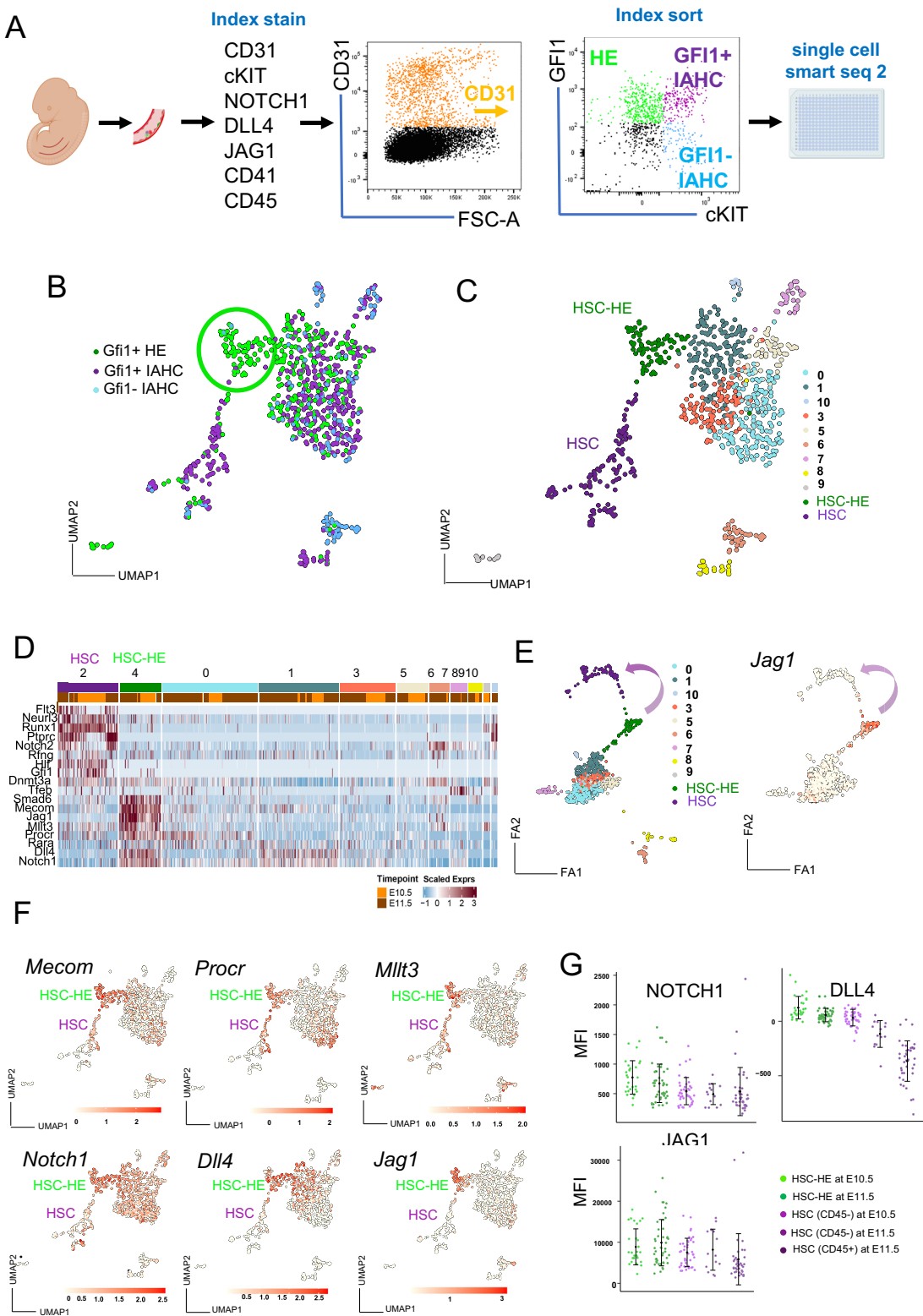

**Fig. 2 | single cell RNA seq of Index sorted HE and IAHC identify HSC-HE and T2-HSC. A** Scheme of the experimental set up. E10.5 and E11.5 AGM lysates were stained for the indicated cell surface markers and Index sorted for Smart seq 2 single cell RNA sequencing. The representative Flow cytometry plot highlights the sorted populations. CD31 + GFI1+cKIT- (GFI1 + HE), CD31 + GFI1+cKIT+ (GFI1 + IAHC) and CD31 + GFI1-cKIT+ (GFI1-IAHC). Cells were sorted in 2 independent experiments from a pool of 5 and 6 at E11.5 and 7 at E10.5 AGMs. **B** UMAP representation of all Index sorted and sequenced cells after quality control ($n$ = 775 cells). The colors highlight the gate they were sorted with GFI1 + HE (green, $n$ = 281 cells), GFI1 + IAHC (magenta, $n$ = 356 cells) and GFI1-IAHC (blue, $n$ = 138 cells). Green circle highlights the cluster of cells that consists only of cells sorted as GFI1-HE. **C** UMAP showing the 11 identified clusters. HSC-HE ($n$ = 76 cells) and HSC ($n$ = 109 cells) are designated. **D** Heatmap of selected HE and HSC specific genes across all clusters.

**E** Forced directed graph layouts of all sequenced cells with colors highlighting individual clusters (left) and *Jag1* normalized gene expression levels (right). **F** UMAPs showing the normalized expression levels of selected genes with annotation of HSC-HE and HSC (**G**) Dot plots showing the Fluorescence Activated Cell Sorting (FACS) Index levels (Mean Fluorescence Level, MFI) for NOTCH1, DLL4 and JAG1 considering all HSC-HE cells or HSC CD45+ or CD45- cells and distinguishing by developmental stage E10.5 or E11.5 (HSC-HE E10.5 $n$ = 29; HSC-HE E11.5 $n$ = 47; HSC E10.5 CD45- $n$ = 43; HSC E11.5 CD45- $n$ = 14; HSC E11.5 CD45+ $n$ = 40). Group HSC E11.5 CD45+ was dropped since it only had three cells. Cells without CD45 MFI information were discarded ($n$ = 9). Cells were labelled as CD45+ for those MFI values greater than Q1 in HSC E11.5 cells. Black dots indicate mean values and error bars refer to +/- standard deviation. (UMAP: Uniform Manifold Approximation and Projection; FA: Force Atlas).

maintain NOTCH1 and JAG1 co-expression on the surface as opposed to CD45 negative CD31+cKIT+GFI1+ subpopulation which loses this co-expression over the course of E10.25 to E11.5 (Fig. 1 G, H, Supplementary Fig. 3C).

### Single cell RNA sequencing identifies different subpopulations of GFI1 + HE

To further understand the identity of Notch receptor and ligand expressing AGM cells and their trajectory, we performed Index sorting (whereby the FACS data of every sorted cell is recorded) with a panel of hematopoietic markers and Notch-related antibodies combined with single cell RNA sequencing using the *Gfi1:tomato* transgenic embryos (Fig. 2A). This approach allows us to link the surface expression of specific Notch signalling molecules to a given cell fate. We index sorted GFI1 + HE (CD31+cKIT-GFI1 + ) (green), HSC containing IAHC (CD31+cKIT+GFI1 + ) (purple) and other HPCs (CD31+cKIT+GFI1-) (blue) from E10.5 and E11.5 AGMs (Fig. 2A, Supplementary Fig. 4*)*. A UMAP plot showed how cells from the different index sorted populations overlapped to a large extent except for a subset of GFI1 + HE cells which were not mixed at all (Fig. 2B, highlighted with green circle). This subset was mainly obtained from both time points (Supplementary Fig. 5A) and were potentially in G1 phase (Supplementary Fig. 5B).

Clustering analysis of all sequenced cells identified 11 cell clusters (Fig. 2C–E) and we obtained the cluster specific marker genes (Supplementary Data 2). The top 25 marker genes from clusters 0, 1, 3 and 5 confirmed a high molecular similarity among them, also observed in the UMAP (Fig. 2B, C and Supplementary Fig. 5C). We then assigned cell identity based on already established [18,25,28] marker gene expression for HE and HSC (Fig. 2D) and assigned this phenotype to clusters 4 and 2, respectively (Fig. 2C–E).

Next, we assessed the molecular relationship between these clusters by trajectory and pseudo-time analysis. We determined that cluster 4 (the pure GFI1 + HE cluster highlighted in green in Fig. 2B) was temporally situated preceding the main two clusters (Fig. 2E and Supplementary Fig. 5D). To our interest, we detected stronger *Jag1* expression levels (Fig. 2E) and increasing levels of key HSC markers, including *Mecom, Mllt3* and *Procr* (Fig. 2F) from cluster 4 toward the more distal HSC cluster (Fig. 2C–F). We therefore will refer to this subpopulation of GFI1 + HE as HSC-primed HE (HSC-HE). Within the distal HSC cluster (cluster 2, mostly composed of GFI1 + IAHC cells) (Fig. 2B), we detected T2 HSC associated marker gene expression (Fig. 2D; Supplementary Fig. 5E) and *Flt3*, a recently identified marker for embryonic multi-potent progenitors (Supplementary Fig. 5F)[8]. Subsequently, we analysed the specific Notch distribution in these AGM subpopulations. We found that *Notch1, Dll4* and *Jag1* gene expression was highly abundant in the HSC- HE clusters with *Jag1* being distinct for this cluster (Fig. 2E, F), suggesting that this ligand has an important role in HSC biology. Taking advantage of the index FACS data, we further confirmed that JAG1 and NOTCH1 proteins were also present at the surface of the cells in the HSC-HE and HSC clusters (Fig. 2G; Supplementary Fig. 5G).

### High *Notch* transcriptional activity identifies a HSC-primed HE cluster and decreases with HSPC maturation

To understand the Notch activity dynamics, we plotted the expression of direct Notch targets across the UMAP (Fig. 3A green circle highlighting the HSC-HE). We observed heterogeneous *Hes1* and *Gata2* expression among the different clusters, but *Hey1*, and specially *Hey2*, were mainly restricted to the HSC-HE cluster (Fig. 3A, HSC-HE highlighted in green). We therefore examined the molecular differences between GFI1 + HE that were co-expressing *Hey1/2* (HSC-HE) and compared it to the remaining cells (excluding *Hey1* and *Hey2* single positive) of GFI1 + HE at the different time points. Besides expressing *Hey1* and *Hey2*, we observed an upregulation of *Notch1* and *Jag1* in both time points (Fig. 3B–D). Differential gene expression analysis (DEA) indicated that several HSC-associated genes were exclusively up regulated in the *Hey1/2* positive (HSC-) HE at E10.5 and E11.5 (Fig. 3B–D, Supplementary Data 3). KEGG pathway overrepresentation analysis further identified gene sets that are known for their involvement in HSC emergence, i.e., signalling pathways in pluripotency in stem cells, HIF1, shear stress, and TNF−signalling (Fig. 3E, Supplementary Data 3). Interestingly, Notch signalling pathway was significantly over-represented at both E10.5 and E11.5 time points in the *Hey1/2* positive (HSC-) HE cluster (Fig. 3E, Supplementary Data 3).

To determine Notch activity dynamics, we compared Notch-target gene expression in the HSC-HE and the HSC cluster. We observed a significant downregulation of *Hey1*, *Hey2* and *Hes1*, while *Gata2* expression was maintained (Fig. 3F). The persistence of the JAG1 and NOTCH1 protein from HE to HSCs (Figs. 1G, H and 2G) but the gradual decrease of Notch target genes from the HE to T2-HSCs (Fig. 3A, F) led us to the hypothesis that JAG1 was not participating in Notch receptor activation (where it would be rapidly endocytosed), but in an alternate function.

### JAG1 and NOTCH1 interactions accumulate in IAHC

Next, we performed IHC to validate our FACS Index distribution of JAG1 and DLL4 in GFI1 + AGM sections. Indeed, we detected both DLL4 and JAG1 in GFI1 + IAHC clusters. Interestingly, we discovered differences in the spatial distribution of these two ligands (Supplementary Fig. 6A). DLL4 was present as discreet foci, whereas JAG1 expressed diffuse across the whole cell surface of GFI1+ cells (Supplementary Fig. 6A). This observation prompted us to study the interactions between the NOTCH1 and the two ligands DLL4 and JAG1 using Proximity Ligation Assay (PLA). In this assay, each point of interaction between a receptor and its ligand is visualized as a fluorescent foci/ signal. Here, DLL4, NOTCH1 and JAG1 antibodies are labelled with oligonucleotides and can serve as primers for rolling circle amplification only if they are close enough to interact (< 30 nm). We multiplexed the PLA for two sets of antibodies (and compatible oligos). One set of PLA probes/antibodies target the extracellular (N-terminal) domains of the NOTCH1 (NOTCH1-extra) and DLL4 (DLL4-extra) and produces a signal in yellow (Fig. 4A, yellow). The second pair of antibodies/probes binds the extracellular (N-terminal) domains of the NOTCH1 (NOTCH1-extra)

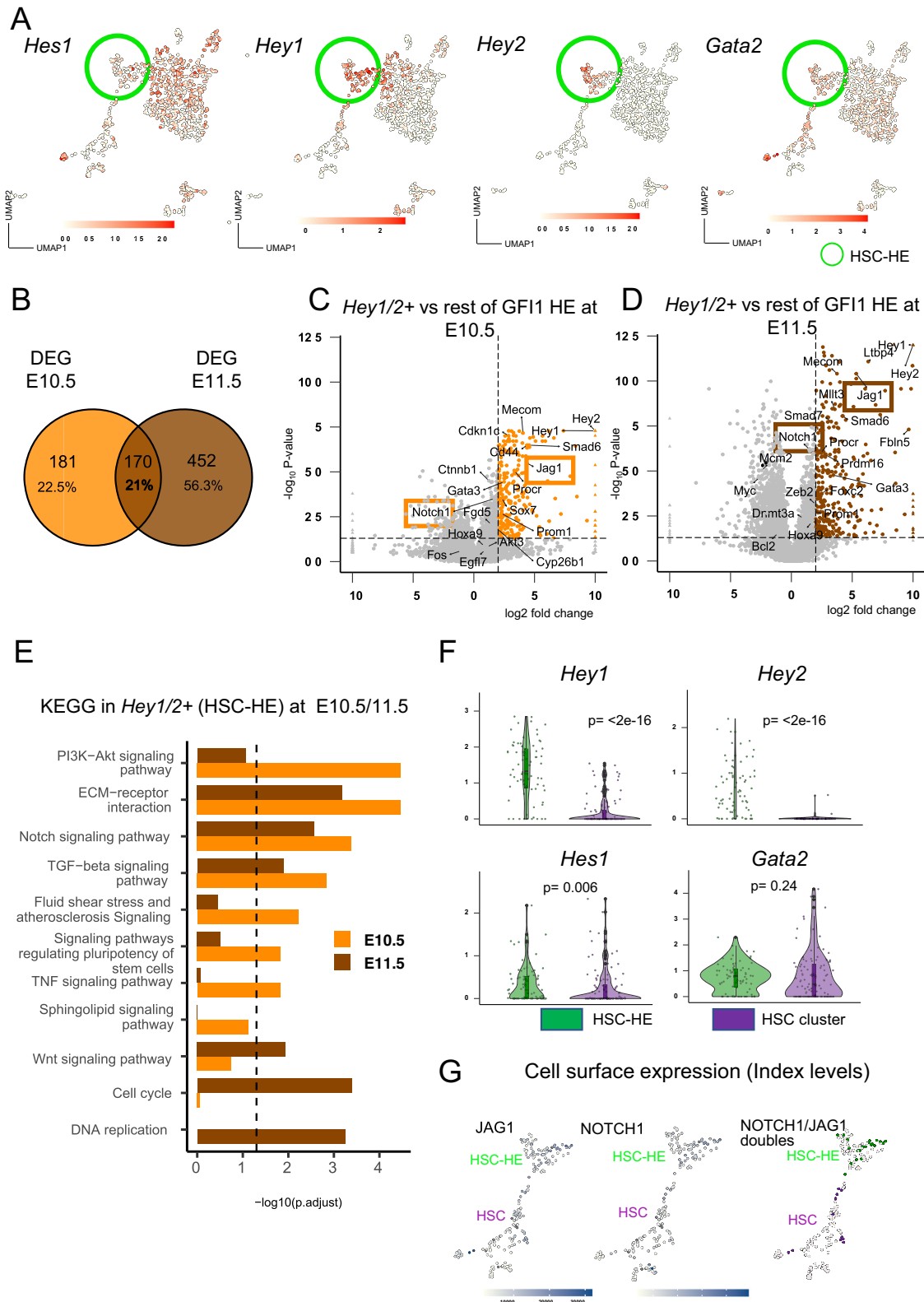

and JAG1 (JAG1-extra) with interactions producing a fluorescent signal in far red (Fig. 4A, magenta). These two sets were used to probe thick (150um) AGM sections of E10.5-E11.5 embryos and imaged as serial confocal images (z-stacks). The 3D rendering of the z-stacks shows punctuated staining for both NOTCH1-extra/DLL4-extra (yellow) and NOTCH1-extra/JAG1-extra (magenta) as expected for this type of assay (Supplementary Fig. 6B, C). For NOTCH1-extra/DLL4-extra interactions

(yellow), we saw a low distribution of fluorescence in endothelial cells and the surrounding tissue, with increased dots also detected in IAHC (Fig. 4B, left and right). Similarly, we detected sparse NOTCH1-extra/ JAG1-extra (magenta) in all AGM tissues, but a much clearer and greater level of signal than for NOTCH1-extra/DLL4-extra in IAHC (Fig. 4B, middle and right). We quantified the total number of amplification dots for each PLA pair in endothelial cells (control) and compared it to

**Fig. 3 | *Hey1/2* expression in HE marks the onset of the HSC specific gene program. A** UMAPs showing the normalized expression levels of *Hes1*, *Hey1*, *Hey2* and *Gata2* with green circle highlighting the HSC-HE population. **B** Venn diagram of differentially expressed genes (DEGs) in GFI1 + HE population when comparing cells expressing *Hey1/2* compared to those not expressing neither of *Hey1/2* at E10.5 (Hey1/2 expressing *n* = 23 and non-expressing *n* = 47) and E11.5 (Hey1/2 expressing n = 36 and non-expressing *n* = 69). DEGs were called with adjusted *p*-value (FDR) < 0.05 and absolute log$_2$ Fold Change > 2. A total of 351 DEGs were found in E10.5 and 622 DEGs in E11.5 (**C, D**) Volcano plots from differential expression analysis between GFI1 + HE *Hey1/2* expressing cells compared to the GFI1 + HE *Hey1/2* non-expressing cells at E10.5 and at E11.5 respectively. A two-sided Wilcoxon rank-sum test was conducted. Obtained *p*-values were adjusted for multiple testing with Benjamini-Hochberg procedure (FDR). Upregulated DEGs are highlighted in orange (E10.5) or brown (E11.5). *Jag1* and *Notch1* expression are pointed out with a box.

Genes showing absolute log$_2$ Fold Change >10 are plotted as triangles (**E**) KEGG Pathways overrepresentation analysis over DEGs of GFI1 + HE *Hey1/2* expressing cells compared to the GFI1 + HE *Hey1/2* non-expressing cells at E10.5 (orange) and E11.5 (brown). A one-sided hypergeometric test was conducted. Obtained *p*-values were adjusted for multiple testing with Benjamini-Hochberg procedure (FDR). Vertical dashed line indicate an adjusted *p*-value of 0.05. **F** Violin plots and boxplots of gene expression levels of *Hes1*, *Hey1*, *Hey2* and *Gata2* comparing HSC-HE (*n* = 76 cells) and HSCs (*n* = 109 cells). Boxplots show the median (centre line) first and third quartiles (box limits), and a maximum of 1.5x the interquartile range (whiskers). Statistical significance was calculated with a two-sided Wilcoxon test. **G** UMAP representation of NOTCH1 and JAG1 co-expressing cells (protein) derived from the index label with HSC-HE and HSC. (UMAP: Uniform Manifold Approximation and Projection; DEG: differentially expressed genes; KEGG: Kyoto Encyclopedia of Genes and Genomes; p: p-value).

the number of dots detected in IAHC. We did not see any significant difference in the fluorescence accumulation for NOTCH1-extra/DLL4-extra (yellow) or NOTCH1-extra/JAG1-extra (magenta) in endothelial cells, but we established significantly higher levels of NOTCH1-extra/JAG1-extra than NOTCH1-extra/DLL4-extra in IAHC (Fig. 4C).

To validate the specificity of the PLA for Notch interactions, we did a control experiment with cells overexpressing *Manic fringe* (MFNG). MFNG is a glycosyltransferase that enhances the binding of DLL4 to NOTCH1. We obtained *Tie2:Mfng* overexpressing AGMs and performed the PLA for both antibody pairs. As expected, and further confirming the validity of the assay, we detected several enhanced interactions for NOTCH1-extra/DLL4-extra (yellow) in IAHC whereas NOTCH1-extra/JAG1-extra (magenta) did not change (Supplementary Fig. 6D).

Curiously, and in agreement with our IHC staining (Supplementary Fig. 6A), most interactions between NOTCH1-extra/DLL4-extra (yellow) in IAHC were located between cells (Fig. 4B white arrows and Supplementary Fig. 6C white arrows), indicating a possible *trans* interaction between adjacent cells. On the contrary, NOTCH1-extra/JAG1-extra (magenta) interactions were frequently detected as interactions that cover the entirety of the cell surface in IAHC, even in the absence of neighboring cells (Fig. 4B, middle and right and Supplementary Fig. 6C). Consistent with FACS and index sorting data that shows NOTCH1 and JAG1 on the surface of the same cell (Figs. 1H, 2F and G, 3G), and the interaction pattern of NOTCH1 and JAG1 by PLA, raised the possibility that NOTCH1 and JAG1 might be interacting on the surface of the same cell (*cis*). This mode of interaction might shield the IAHC from further NOTCH1 interaction with ligands presented from the surrounding cells.

### JAG1 and NOTCH1 interact in *cis* in IAHC

To test this hypothesis, we probed the intracellular domains (C-terminus) of the NOTCH1 and JAG1 with the rationale that we would only detect a signal if these intracellular parts were on the same cell, in parallel orientation to each other and therefore close enough to interact (*cis* configuration) (Fig. 4D). We multiplexed the new antibody pair, NOTCH1-int/JAG1-int (green) with the two previous probe pairs and analysed their fluorescence distribution in the AGM sections. Most importantly, we quantified the number of interactions from NOTCH1-extra/JAG1-extra (magenta) and NOTCH1-int/JAG1-int (green) in IAHC. In agreement with our hypothesis, we detected accumulation of the green NOTCH1-int/JAG1-int signal specifically in cKIT+ cells/ IAHC (Fig. 4E; Supplementary Fig. 6E). Intriguingly, we did not find a significant difference in the number of interactions for NOTCH1-extra/JAG1-extra (magenta) and NOTCH1-int/JAG1-int (green) (Fig. 4F), further suggesting that most NOTCH1-JAG1 interactions in IAHCs are *cis*-interactions (detected with both intracellular and extracellular PLA).The capacity of a ligand-expressing cell to *cis*-inhibit the NOTCH receptor depends on the concentration of intracellular and extracellular ligand[35]. Thus, we tested whether *cis* interactions between

NOTCH1 and JAG1 in IAHC can be interrupted by supplying JAG1 in excess (in *trans*) (Fig. 4G). We treated E10.5 AGM as explants for 4 h in the presence of IgG or soluble, recombinant JAG1 (Fc-JAG1) and performed PLA for NOTCH1-extra/JAG1-extra (magenta) and NOTCH1-int/JAG1-int (green) as previous (Fig. 4G, H). In these in vitro conditions, IgG treated explants showed reduced number of NOTCH1-int/JAG1-int interactions compared to untreated IAHC (compare Fig. 4E, F with Fig. 4H, I), however exposure to exogenous Fc-JAG1 further decreased the number of NOTCH1-int/JAG1-int interactions (Fig. 4H, I), strongly suggesting a loss of *cis* interaction between NOTCH1 and JAG1 upon Fc-JAG1 treatment. To substantiate our hypothesis further, we undertook two additional approaches. In the first instance, we treated wild type AGMs as explants with Fc-JAG1, anti-JAG1, anti-NOTCH1, or combined Fc-JAG1 with anti-NOTCH1 for 4 h and collected CD31+cKIT+CD45+ cells for notch targets gene expression profiling (Fig. 5A, Supplementary Data 1). In agreement with our observations that the *cis*-interaction were reduced upon Fc-JAG1 treatment in PLA assays, we detected higher levels of Hes1 and Gata2 when treated with Fc-JAG1 or anti-JAG1 (Fig. 5A). Next, we induced the genetic deletion of Jag1 in endothelial and hematopoietic cells by using a Ve-cadherin-CreERT2/Jag1floxed mouse line. We induced the deletion with 10uM of 4-OHT starting from E10.5 ex vivo and subsequently profiled the CD31+cKIT+CD45+ cells for notch targets gene expression (Fig. 5B, i). As in the previous experiments, the levels of Hes1 and Gata2 were elevated in Jag1 deficient cells (Figure 5Bii). In both approaches, we also included anti-NOTCH1 with Fc-JAG1 or Jag1 deletion. In both instances, the higher levels of Hes1 and Gata2 in Fc-JAG1 or Jag1 deleted cells was reverted linking NOTCH1 as the receptor for the identified cis-interaction (Fig. 5Aii, Bii).

### *Cis* interaction between JAG1 and NOTCH1 reduces Notch activation in IAHC

Based on our previous findings, we speculated that if the role of JAG1 in IAHC was to shield them from Notch signaling activation, then the IAHC should be less responsive to Notch activation than cells without *cis*-inhibition, ie endothelial cells. We therefore purified endothelial cells (CD31+cKIT-GFI1-) as a reference and IAHC (CD31+cKIT+GFI1 + ) in parallel from E11.5 AGMs (Fig. 5A). Both populations of cells were incubated overnight with the γ-secretase inhibitor, compound E (CompE), to abolish basal Notch activity. Both populations were subject to different conditions for 4 h after which the cells were harvested for gene expression analysis (Fig. 6A). The cells were either kept in a Notch inhibited state (compE), released out of Notch inhibition (wash), or stimulated with Fc-JAG1 (Fig. 6A) to activate Notch signaling in *trans* while disrupting *cis*-inhibition. To minimize cell-cell interactions, cells were seeded at low density (300-400 cells/well) in semi-solid/semi-rigid methylcellulose (with growth factors). We supplemented the methylcellulose culture with a nascent RNA capture reagent (EU) to capture and quantify the transcriptional changes that occur in each condition and population. We first validated this experimental

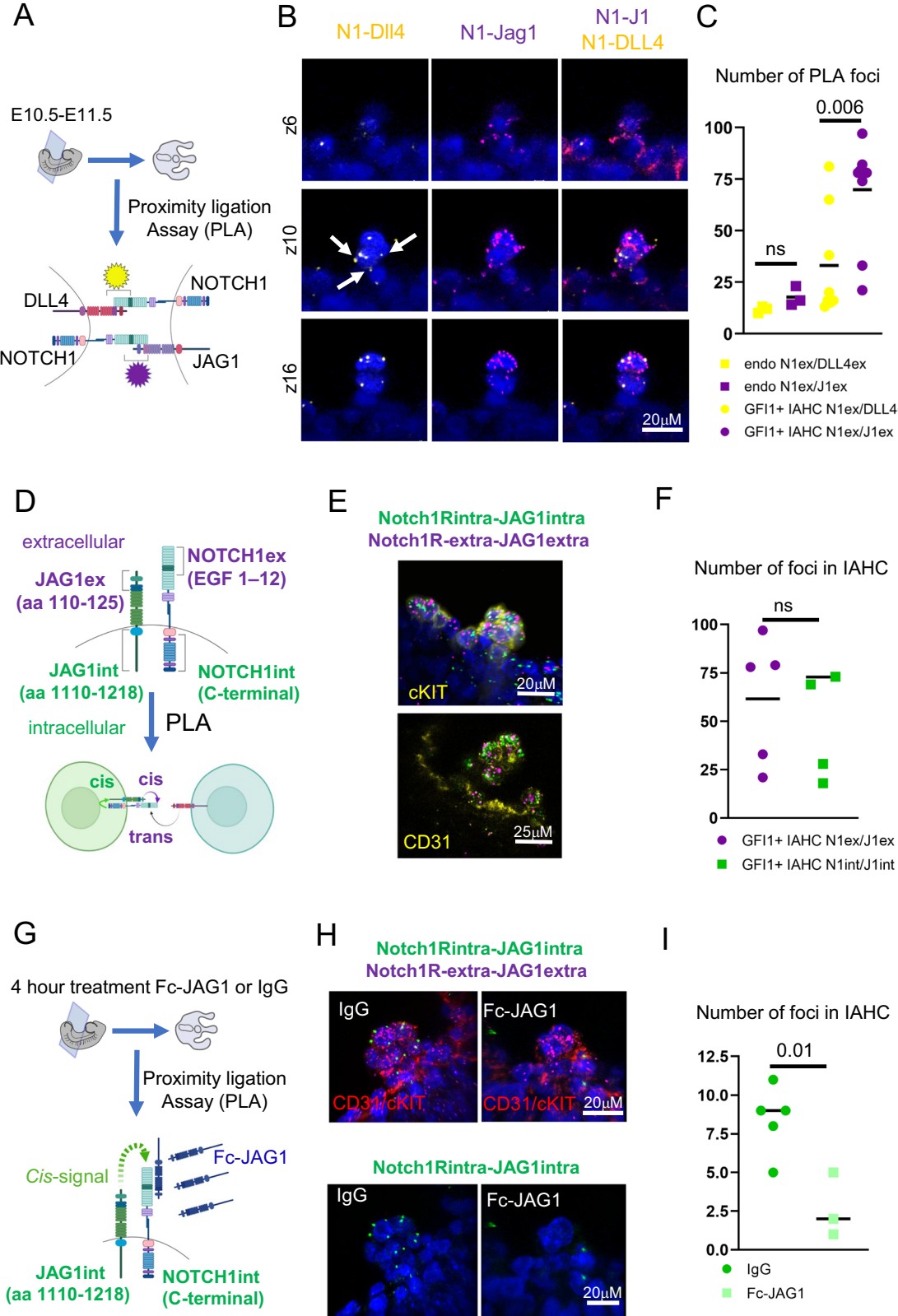

approach by performing qPCRs for the Notch targets *Hes1* and *Gata2* since they are both expressed in endothelial cells and HSCs (Fig. 3A). We detected lower levels of both in the compE condition and higher levels with Fc-JAG1 stimulation (Fig. 6B), confirming that excess of Fc-JAG1 was activating NOTCH1. However, the up-regulation of *Hes1* and *Gata2* in response to Fc-JAG1 was more robust in endothelial cells than in GFI1 + IAHC cells (Fig. 6B), indicating that GFI1 + IAHC cells are more

refractory to Notch activation. These results are compatible with a *cis*-inhibitory function of JAG1 specifically in GFI1 + IAHC cells.

### NOTCH1-JAG1 interactions in *cis* inhibit lineage differentiation genes

To further understand the effects of Notch manipulation in nascent transcription, we proceeded to sequencing the nascent RNA of the

**Fig. 4 | NOTCH1 and JAG1 form *cis* interaction in IAHC. A** Scheme of experimental set up. 150μm thick trunk sections of E11.5 AGMs were subject to Proximity ligation assay (PLA) with indicated antibody pairs followed by confocal imaging. **B** Representative optical 2–3 μm sections (z-stacks,z) through a IAHC. PLA signals are detected as spots for points of interactions. NOTCH1/DLL4 (yellow), NOTCH1-ext/JAG1ext (magenta) and DAPI. Scale = 10 μm. **C** Quantification of foci (interaction points) for NOTCH1/DLL4 (yellow), NOTCH1ext/JAG1ext (magenta) in endothelial cells (endo, $n = 3$) or IAHC ($n = 7$). Statistical significance was calculated with two-tailed t-tests from 4 independent experiments). Vertical error bars indicate the mean and standard deviation values. Source data are provided as a source data file. **D** Scheme of experimental set up to distinguish between *trans* and *cis* NOTCH1-JAG1 interactions with antibodies recognizing the indicated amino acid sequence of NOTCH1 or JAG1. NOTCH1ext/JAG1ext (magenta) and NOTCH1int/JAG1int (green).

**E** Exemplary images of a cKIT+ or CD31 + IAHC probed with PLA with NOTCH1ext/JAG1ext (magenta) and NOTCH1int/JAG1int (green) and DAPI. **F** Quantification of foci (interaction points) for NOTCH1ext/JAG1ext (magenta) and NOTCH1int/JAG1int (green) in IAHC ($n = 5$). Statistical significance was calculated with two-tailed t-tests in 3 independent experiments. Vertical error bars indicate the mean and standard deviation values. Source data are provided as a source data file. **G** Experimental set up to test if NOTCH1-JAG1 *cis* interactions can be disrupted with Fc-JAG1 in *trans*. **H** Representative images of a cKIT+ and CD31 + IAHC probed with PLA for NOTCH1ext/JAG1ext (magenta) and NOTCH1int/JAG1int (green) and DAPI. **I** Quantification of foci (interaction points) of NOTCH1int/JAG1int (green) in IAHC. Statistical significance was calculated with two-tailed t-tests ($n = 8$ individual IAHC from pools of 7 embryos per condition). Vertical error bars indicate the mean and standard deviation values. Source data are provided as a source data file.

GFI1 + IAHC samples after Notch signaling manipulation (Supplementary Fig. 6F). Differentially expressed genes (DEGs) were independently identified for compE and Fc-JAG1 conditions relative to the washout, which was considered as the reference (Fig. 6C). Those DEGs in the compE scenarios (962 genes) were classified as classical Notch responders (Fig. 6D, Supplementary Data 4). Part of them (384) were in common with those obtained from Fc-JAG1 comparison. Finally, a total of 1,724 genes were exclusively differentially expressed when Fc-JAG1 was added to the culture. From these, the set of up-regulated genes encompassed genes associated to mitotic cell cycle transition, myeloid and lymphoid cell differentiation, chromatin remodeling and autophagy, based on the overrepresentation analysis of GO BP database (Fig. 6E, Supplementary Data 4). The enrichment of these terms suggests that disrupting the *cis* inhibition (NOTCH1-JAG1) by *trans* activation through Fc-JAG1 triggers cell cycle entry and activation of lineage differentiation program. We further identified a more naïve state of the GFI1 + IAHC before incubation with Fc-JAG1 by examining genes involved in MHC class I antigen presentation/processing and Immune response since naïve HSCs typically show lower abundance of these group of genes[47,48]. We found several genes of these two processes significantly up-regulated upon Fc-JAG1 stimulation (Fig. 6F and Supplementary Data 4). Finally, Notch signaling-related molecules were also up-regulated upon stimulation with Fc-JAG1.

In summary, we find evidence that within the GFI1 + IAHC cells that retain NOTCH1 and JAG1 in *cis* conformation, Fc-JAG1 causes activation of genes associated with cell cycle entry and differentiation. These results agree with our hypothesis that the function of the *cis* conformation is to preserve a naïve HSC state within the GFI1 + IAHC cells.

### NOTCH1-JAG1 interactions in *cis* is reduced upon RFNG knockdown

Notch post-translational modifications by FRINGE glycosyltransferases can enhance ligand-receptor interactions in a specific way. *Rfng* expression was detected in the T2-HSC population (Fig. 2D). RFNG has been reported to enhance NOTCH1-JAG1 *cis* interactions in cultured cells[34]. To test whether the *cis* interaction in HSCs could be modulated by RFNG, we performed IHC for RFNG on *Gfi1:tomato* AGMs. We detected RFNG positive cells in a few GFI1 + IAHC of E11.5 AGMs (8 out of 19, Fig. 7A). To quantify the RFNG positive cells within the whole AGM, we performed FACS analysis, combined with the HSC markers, Sca1 and EPCR (T2-HSCs)[25]. We detected significant accumulation of RFNG positive cells in T2-HSCs and some EPCR positive cells, but not in Sca1-EPCR- IAHCs (Fig. 7B, C; Supplementary Fig. 7A). We went further and sub-gated the T2-HSC fraction for RFNG + /v cells to determine if these cells were co-expressing NOTCH1 and JAG1 in the *cis* confirmation. In support of our hypothesis, RFNG positive T2-HSCs enriched for NOTCH1 and JAG1 co-expression (Fig. 7D, E).

Finally, we performed knock down experiments of RFNG in ex vivo AGM cultures with anti-*Rfng* FANA-ASO nucleotides (Fig. 7F). These nucleotides enter the cell, bind to the transcript and recruit

RNaseH to degrade the mRNA (Supplementary Fig. 7B). We cultured control (scramble FANA-ASO) or *Rfng* ASO treated AGMs for two days and analysed the T2-HSC and NOTCH1-JAG1 co-expression frequency by FACS (Fig. 7F; Supplementary Fig. 7C).

In agreement with our hypothesis, a decrease in RFNG in the AGM was observed concurrently with significantly fewer phenotypic HSCs (CD45 + CKIT + SCA1 + EPCR + ) and a reduction in NOTCH1-JAG1 co-expression (Fig. 7F, G, Supplementary Fig. 7C).

Altogether these results indicate that RFNG might preserve the NOTCH1-JAG1 *cis* interaction in some IAHC cells and maintain their HSC phenotype.

## Discussion

Notch signalling is required for a plethora of cellular decisions that control cell states and cell fate acquisition, including hematopoiesis. In the embryonic aorta, hemogenic endothelial cells give rise to HSPCs in close association with the aortic endothelium. Arterial and hemogenic endothelium and HSPCs are specified in the AGM region, and all require a specific level of Notch activity. Earlier studies have elucidated that NOTCH1, JAG1, RBPJ and HES repressors are key for the generation of HSPC[42,49–52], while NOTCH1, NOTCH4, DLL4 and RBPJ are required for arterial development[53–56]. Therefore, it is concluded that Notch activity is compulsory for HSC emergence and arterial formation, but the timing and the threshold of Notch activity as well as the precise interactions of receptors and ligands remained undetermined. In the AGM, blocking DLL4 or inhibiting Notch activity with γ-secretase inhibitors can increase HSPC activity[17,57,58], yet absence of JAG1 leads to a dramatic loss of IAHC[44,45]. Importantly, the phenotypes of blocking DLL4 with an antibody or treatment with γ-secretase inhibitors is dependent on the developmental stage of HE and HSPC; blocking DLL4 ex vivo most efficiently increases HSC activity when applied at early AGM stages (31–34 s)[17]. Likewise, blocking of Notch activity with γ-secretase inhibitors before acquiring HSC fate can diminish HSC activity[46]. Moreover, Notch1 receptor levels are also tightly regulated during HE and EHT by Sox17 and Gpr183[57,59].

Taken together, we conclude that Notch activity needs to be dampened once the (HSC)-HE has been specified. In agreement with this assumption, Notch activity tracing lines detect lower notch activation in IAHC than their arterial surrounding[43,46,57,59]. Here, we present evidence for a *cis* interaction between NOTCH1 and JAG1 that is especially relevant in T2-HSCs and modulated by RFNG that integrates and further explains the previous observations. Increasing JAG1 levels on the cell surface blocks Notch signalling by forming a *cis* (inhibitory) conformation with NOTCH1. A previous study already established higher levels of activated NOTCH1 (NICD) in IAHC in the absence of JAG1. The *Jag1* deficient cKIT+ IAHCs were present in the aorta but could not bud into the lumen of the aorta, and instead stayed embedded in the endothelium and expressed high levels of endothelial specific genes, indicating that hematopoietic maturation was blocked[43]. Similarly, and in support of the opposing effect for DLL4 and JAG1 in HSC specification, AGMs treated with a DLL4 blocking

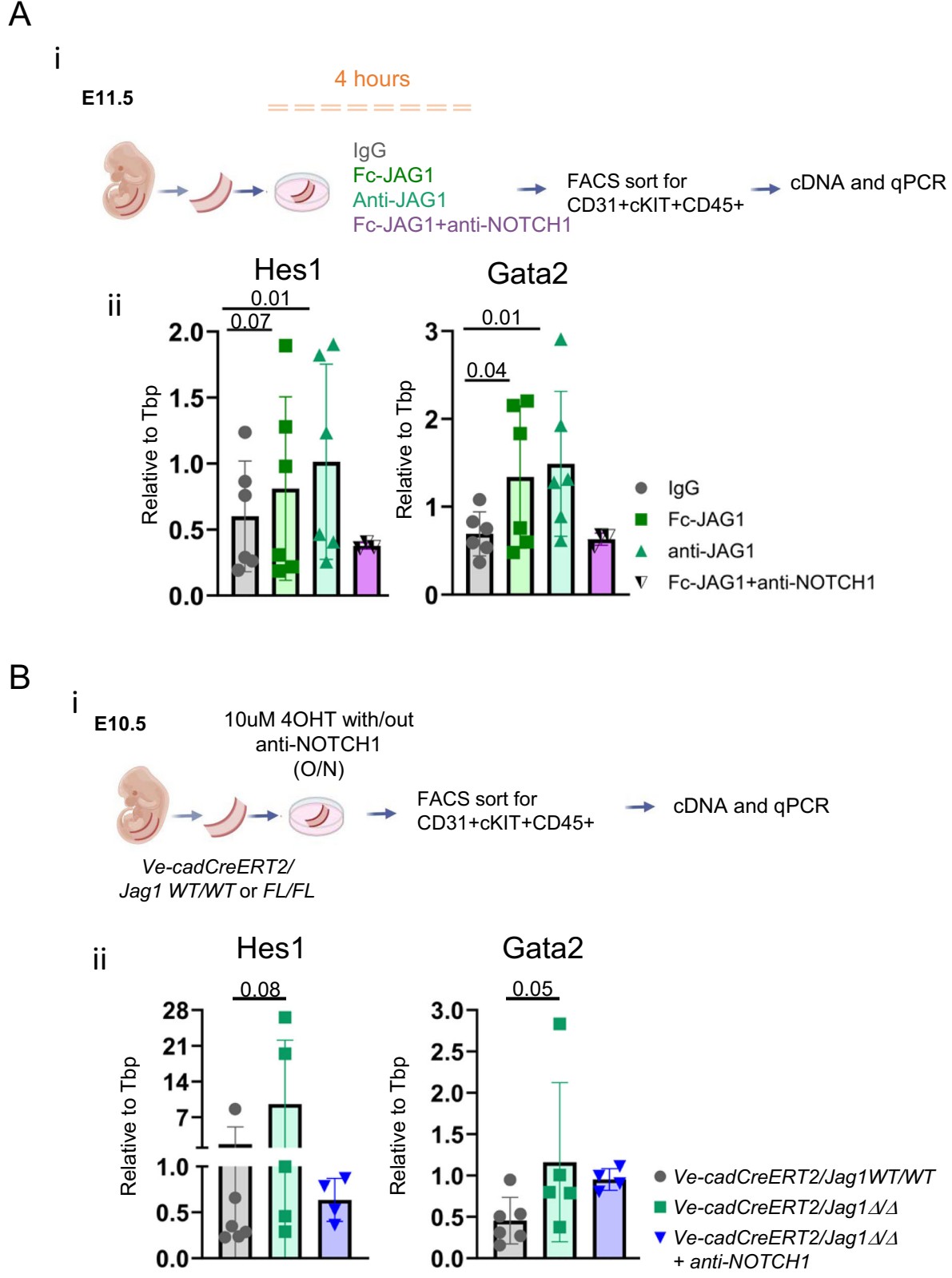

antibody show enhanced HSC activity and reduced target gene expression[17].

The time course of Notch ligand and receptor expression in the hemogenic and IAHC subpopulations suggests that this *cis* interaction is specifically maintained in T2-HSC, while other HPC lose this co-expression. We speculate that a fine balance of Notch activity that is most likely induced by *trans* activation by DLL4 and increasing

*cis* inhibition through JAG1 are key for permissive conditions of HSC fate. DLL4 activation should be inhibited since it has a detrimental effect over HSC activity[43], however *cis* inhibition could also prevent NOTCH1 from responding to free JAG1 or other expressed ligands such as JAG2[49]. We show that the persistence of this *cis* NOTCH1-JAG1 is linked to RFNG expression, a glycosyltransferase that modifies the NOTCH1 receptor to have a stronger affinity for JAG1 in *cis*[34,39]. We find

**Fig. 5 | JAG1 perturbation in CD31 + cKIT + CD45 + IAHC increases Hes1 and Gata2 levels. A** (i) Scheme of experimental procedure. E11.5 AGMs were incubated for 4 h with culture media containing IgG (50 ug/ml), Fc-JAG1 (4 ug/ml), anti-JAG1 (50 ug/ml), anti-NOTCH1 (50 ug/ml) or Fc-JAG+anti-NOTCH1 (50 ug/ml each). AGMs were then processed for FACS purification of CD31+cKIT+CD45+ cells. After this time, RNA was extracted, cDNA was pre-amplified and subject to qPCRs for Hes1 and Gata2 levels. (ii) Relative gene expression levels of *Hes1* and *Gata2* to Tbp (Tata binding protein) in IgG, Fc-JAG1, anti-JAG1, anti-NOTCH1 and FcJAG1+anti-NOTCH1. Results were obtained from 2 independent experiments with 4 AGMs/experiment (*n* = 12 embryos) and statistical significance was calculated with one-tailed t-tests. Vertical error bars indicate the mean and standard deviation values. Source data are provided as a source data file. **B** (i) Scheme of experimental procedure. E10.5 Ve-cadCreERT2/Jag1 WT/WT or FL/FL AGMs were treated with 10 uM of 4OHT with, or without anti-NOTCH1 overnight. AGMs were then processed for FACS purification of CD31+cKIT+CD45+ cells. RNA was extracted, cDNA was pre-amplified and subject to qPCRs for Hes1 and Gata2 levels. (ii) Relative gene expression levels of *Hes1* and *Gata2* to Tbp (Tata binding protein) in *Ve-cadCreERT2/Jag1 WT/WT*, *Ve-cadCreERT2/Jag1Δ/Δ* and *Ve-cadCreERT2/Jag1Δ/Δ+* anti-NOTCH1. Results were obtained from 2 independent experiments with 2 and 3 AGMs/sample (*n* = 5 embryos) and statistical significance was calculated with one-tailed t-tests. Vertical error bars indicate the mean and standard deviation values. Source data are provided as a source data file. (O/N: overnight).

RFNG concentrated in sparse cells within a few IAHC by IHC and by FACS, we specifically detect them in EPCR + /Sca1 + / T2-HSC. Finally, AGMs after knock down of *Rfng* showed a reduction of NOTCH1-JAG1 co-expressing T2-HSCs. By nascent RNA capture assay, we identified the pathways that are regulated by JAG1 in the HSC population. We find enrichment of gene sets that include cell cycle, chromatin remodeling, lineage differentiation, and antigen processing/presenting genes that are upregulated in response to Fc-JAG1, strongly indicating that the *cis* confirmation of NOTCH1-JAG1 preserves a naïve stem cell fate phenotype. Although the nascent RNA assay has given an indication of the processes affected by NOTCH1-JAG1 interactions, given the inherent noisiness of the samples further experimental validation of specific targets would be needed.

It is worth mentioning that deletion of all three Fng genes has minor effects on adult hematopoiesis[60], however phenotypic and functional studies of Rfng KO embryonic HSC could provide critical insight into HSC specification in the AGM. It is not uncommon that HSC-specific phenotypes are not detected unless studied in serial transplantation settings[61].

It is interesting to note that some HSC specific genes, including *Procr*, *Mecom*, *Fgd5*, and *Hoxa9* are expressed at higher levels in a subset of the GFI1 + HE that has high levels of *Jag1* and some Notch target gene expression (Figs. 2D–F and 3A). This suggests that the HSC fate is established as early as the HE state, maybe through Notch signaling and this is preserved in the IAHC. In addition, we find that these precursors have a low proliferation index to ensure their integrity (Supplementary Fig. 5B). Of note, NOTCH1 mutant AGM explants showed significantly higher levels of BrdU+ incorporation in the hematopoietic compartment[57] suggestive of Notch signalling controlling cell cycle dynamics in the HE/IAHC. The notion that the HE is heterogeneous in regards of their hematopoietic potential has been recently postulated[62,63]. We are therefore further confirming this finding and are proposing a mechanism, namely NOTCH1-JAG1 in *cis*, as the driving force that maintains the HSC fate in the IAHCs.

*Cis* interactions per se have been postulated based on mathematical modelling of Notch receptor and ligand interactions. This model of Notch activity fine tunes tissue pattern induction and is further validated as a vital mode of signalling in engineered cell cultures systems[34–36,64]. *Cis* inhibition has been shown to prevent receptor activation when ligands are in stochiometric access[65] and functionally demonstrated in *Drosophila* development of the wing, eye, oogenesis and the notum[66–68]. Disruption of *cis* inhibition by mutating specific extracellular regions of the *Jag* orthologue, *Serrate*, results in the wing vein loss phenotype[69]. Despite the different examples in *Drosophila*, demonstration of similar *cis*-inhibitory mechanisms in vertebrates has been elusive, likely due to the higher complexity of Notch signaling. However, mathematical modelling of the Notch pathway predicts that *cis* interactions are an integral part of Notch signaling also in vertebrate systems and needs further exploration[37]. This study presents unprecedented phenotypic, functional, and visual evidence of such interaction in the vertebrate tissue.

Finally, generating HSC through re-programming of somatic or differentiated cells, or the in vitro differentiation of (induced Pluripotent) Embryonic Stem (ES) cells is a goal of regenerative medicine that is still unmet. Indeed, several studies have documented the expression and requirement of Notch signaling molecules for inducing a "definitive" blood precursor population[62,70–75]. Still, the repopulation capacity of these cells is poor, lineage biased and/or only transient in the recipient[76–82]. We speculate that the *cis* (inhibitory) conformation might be a vital mode of regulation that is lacking in in vitro systems. Further studies are needed to dissect the modules of Notch signaling in ES cell-based differentiation to blood and try to implement the missing interactions.

## Materials and Methods
### Mouse lines and animal work
The CD1 wild type strain and *Gfi1:tomato* (Thambyrajah et al. [11]) were used in this study. For time matings, *Gfi1:tomatot^{omato}* or CD1 WT females were mated to *Gfi1^{tomato}* or CD1 WT males. The resulting embryos were genotyped and used for FACS analysis and sorting, IHC and Index sorting. Jag1floxed mice (B6.129S-Jag1tm2Grid/SjJ purchased from the Jackson laboratory) were bred to Ve-cadherin CreERT2 mice (Monvoisin et al, 2006). Vaginal plug detection was considered as day 0.5. To induce the deletion of the floxed Jag1 alleles, 10uM of 4 hydroxytamoxifen (Sigma #7904) was added to the explant media. Animals were kept under pathogen-free conditions, and all procedures were approved by the Animal Care Committee of the Parc de Recerca Biomedica de Barcelona, license number 9309 approved by the Generalitat de Catalunya.

### Genotyping PCR
Small pieces of embryonic tissue or yolk sac were dissected off the embryo and placed in PCR tube containing 100 µl of PBS. The tissue pieces were boiled for 8 min at 98 °C for denaturation and further digested with Proteinase K (50 µg/ml) for 30 min at 55 °C, and the enzyme deactivated by boiling the samples for a further 10 min at 95 °C. 1 µl of the samples was used as a template for the PCR.

### AGM dissection, single cell suspension
AGMs of E10 - E11.5 embryos were dissected in PBS with 7% fetal calf serum (FBS) and penicillin/streptomycin (100 U/mL). Single-cell suspensions were generated by incubating the tissues for 20–30 min in 500 ul of 1 mg/ml of Collagenase/Dispase (Roche cat# 10269638001) before mechanical dissociation with a syringe and needle. The resulting single-cell suspension was used for antibody staining (see Supplementary Data 1 for list of antibodies). Samples were analysed on a Fortessa instrument or sorted with FACS AriaII or BD Influx (BD Biosciences). FACS plots were generated using FlowJo V10.

### cDNA generation and qPCRs
FACS purified CD31+cKIT+CD45+ cells were converted into cDNA using the SuperScript™ IV Single Cell/Low Input cDNA PreAmp Kit (ThermoFisher # 11752048) according to Manufacturer's instructions and 13 cycles pre-amplification with random primers. Sybergreen-based qPCRs were performed with primer pairs targeting Tbp, Hes1, Hey1/2 and Gata2 (Supplementary Data 1).

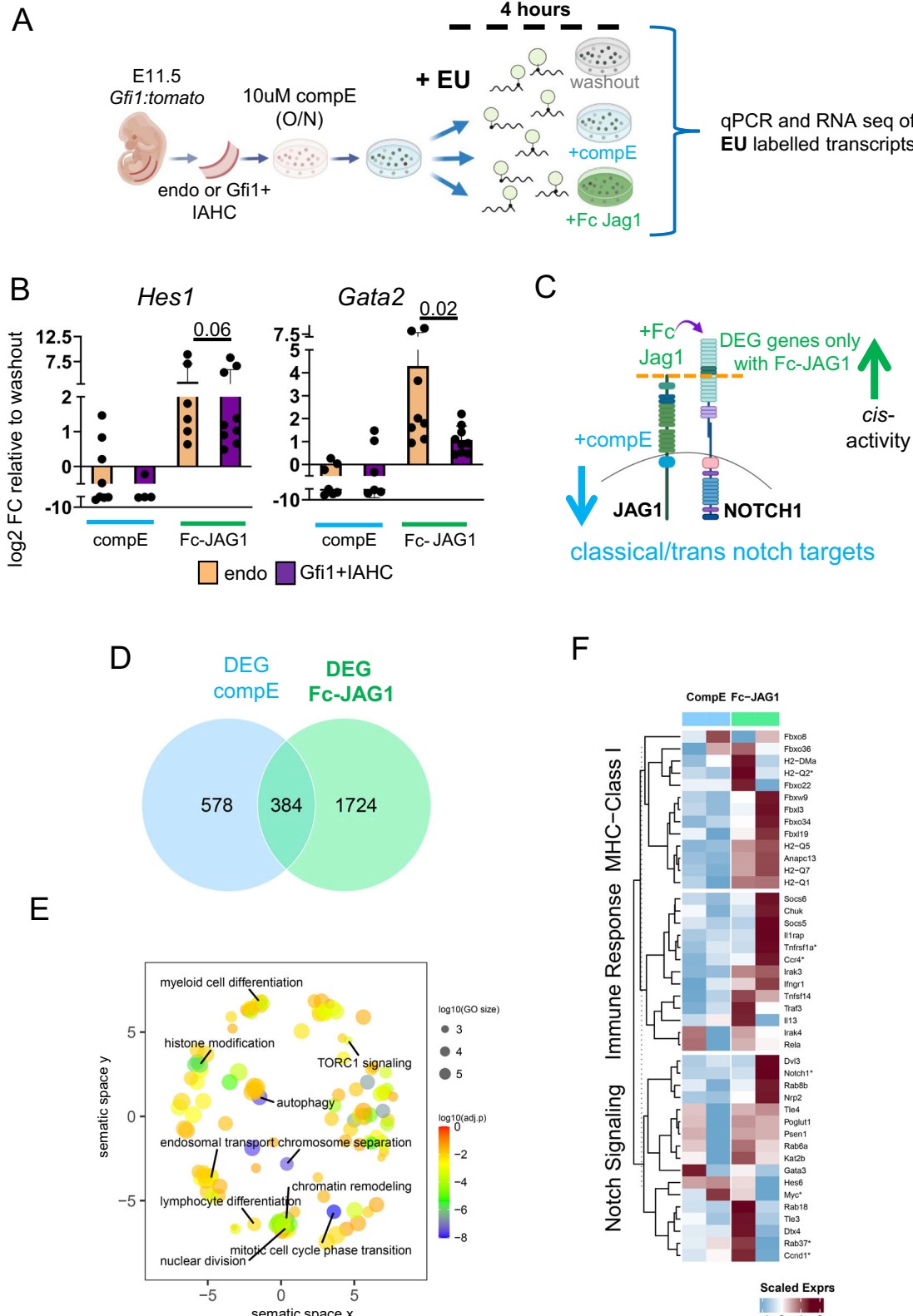

## Single-cell RNA sequencing

E11.5 AGMs were prepared for flow sorting as described above. Single cells were sorted into 384-well plates containing lysis buffer and snap frozen. Libraries were prepared using a modified version of the Smart-Seq2 protocol. Briefly, cDNA was prepared using a Mantis platform (Formulatrix) and quantified with quantIT picogreen reagent (Thermo Fisher). Dual indexed sequencing libraries were prepared from 0.1 ng cDNA using an Echo525 automation system (Labcyte) in miniaturized reaction volumes. The library pool was quantified by qPCR using a Library Quantification Kit for Illumina sequencing platforms (Kapa Biosystems). Paired-end 75 bp sequencing was carried out by clustering 1.5pM of the library pool on a NextSeq 500 sequencer (Illumina).

**Fig. 6 | JAG1 in GFI1 + IAHC controls stemness related pathways. A** Scheme of experimental procedure. Endo (CD31-GFI1-ckit-) and GFI1 + IAHC from E11.5 AGM were FACS purified and treated with 10 µM of compE overnight. The cells were then washed and split into 3 conditions as illustrated. Either cells remained in a washout condition (control), returned to 10µM of CompE treatment (CompE) or cultured with Fc-JAG1. 3 independent experiments with cells purified from pools of embryos (E11.5 AGM $n = 15$ embryos) were performed. All conditions were cultured in semi-solid/semi rigid methylcellulose at low concentration to limit cell-cell contact and 10 µM of EU was added to each culture to label the newly synthesized RNA for 4 h. After this time, RNA was extracted, cDNA was pre-amplified and subject to qPCRs and RNA sequencing. **B** Relative gene expression levels of *Hes1* and *Gata2* in compE and Fc-JAG1 treated samples compared to washout control by qPCR. Results were obtained from 3 independent replicates and statistical significance was calculated with two-tailed t-tests. Vertical error bars indicate the mean and standard deviation values. Source data are provided as a source data file. **C** Illustration of NOTCH1 and JAG1 in a *cis* conformation of a cell and the expected deregulation of notch targets in response to compE treatment (blocks all Notch activity) and Fc-JAG1 (binds to freely available NOTCH1 or activates NOTCH1 by competing with the JAG1 in *cis*). **D** Venn Diagram of differentially expressed genes (DEGs) obtained from the independent comparison of CompE and Fc-JAG1 conditions replicates against the washout condition, considered as the reference group. DEGs were called with adjusted *p*-values (Benjamini-Hochberg procedure, FDR) < 0.05 and expressed in at least two replicates in any of the two conditions being compared. A Wald-test was conducted for each pairwise comparison. **E** Overrepresented GO BP terms (adjusted p-value (Benjamini-Hochberg procedure, FDR) < 0.05) obtained from exclusively DEGs in Fc-JAG1 against washed-out comparison. A one-sided hypergeometric test was conducted. Terms are summarized in their semantic space. **F** Heatmap of representative genes associated to MHC-Class I, Immune Response or Notch signaling pathways. Relative expression levels in CompE and Fc-JAG1 treated samples compared to their paired washed-out condition are shown. Selected genes were DEGs from Fc-JAG1 against washed-out comparison. Genes with an asterisk showed an adjusted p-value (FDR) < 0.1. (O/N: overnight, EU: 5-ethynyluridine).

## scRNASeq data analysis

Raw reads were mapped against the Mus musculus genome (mm10) with STAR aligner tool (v2.7.3)[83]. HTSeq framework (v0.9.1) was used for gene expression quantification[84]. Raw count matrix contained data from 860 sequenced cells and 46,170 genes. Python v3.7.3 and scanpy (v1.4.4) were used for data pre-processing and main downstream analysis. Other relevant Python packages included anndata (v0.6.22.post1), umap (v0.3.10), pandas (v0.25.1), scikit-learn (v0.21.3) and statsmodels (v0.13.15). Cells with less than 2,500 expressed genes and less than 50,000 counts were discarded, and no more than 5% mitochondrial gene expression. Genes expressed in at least one cell were kept. No batch correction was performed. After quality control, we obtained an expression matrix including 775 cells and 30,362 genes. Cyclone was used to classify cells into G1, G2 or S cycle phases[85]. Overall gene expression was normalized to 10,000 counts and logarithmically transformed. Then highly variable genes (HVG) were selected with default parameters. A total of 7,925 HVG were identified. Read depth, mitochondrial gene content and cell cycle effects were regressed out. For cells visualization and clustering, PCA was first performed with 50 components on the list of HVG. UMAP was obtained after neighborhood graph was computed (n_neighbors = 5 and 50 PCs). Cells were clustered using the Louvain algorithm with a resolution of 0.5. This analyisis identified 11 clusters. Cell identities annotation to clusters was based on already established marker genes from the literature. Cluster-specific marker genes were found using the scanpy function rank_genes_groups using Wilcoxon rank-sum test method. Only those genes present in at least 25% of cells in either groups (cluster of interest against the rest) were considered.

Benjamini-Hochberg procedure was used to obtain adjusted p-values. Forced Directed Graph (FDG) was computed after obtaining the diffusion map with default parameters. For the layout, fa2 (v0.3.5) was selected. Cells pseudotime was estimated considering HSC-HE cluster cells as the root.

Data visualization was performed with the ggplot2 (v3.4.1), complexHeatmap (v2.14.0) and EnhancedVolcano (v1.16.0) R packages (v4.2.1)[86]. Differential expression analysis (DEA) was performed for cells simultaneously expressing Hey1 and Hey2 genes from GFI1 + HE population (subset of 281 cells). Genes were considered expressed if their number of normalized counts was higher than 0. DEA was independently conducted per time point (E10.5 and E11.5). Same approach was applied as for the cluster-specific markers identification. Those genes with adjusted p-value (FDR) < 0.05 and absolute log2 FC > 2 were identified as differentially expressed genes (DEGs).

## Immuno- histo-chemistry

E10.5 embryos were fixed in 2% Paraformaldehyde (Thermo Fisher) for 12 min, before they were soaked in 30% sucrose overnight and mounted in OCT compound. 10−12 μm sections were prepared using a cryostat. The sections were permeabilsed at −20 °C for 10 minutes in −20° C 100 Methanol followed by serum blocking (PBS with 3% BSA, 0.5% FCS, 0.05% Tween20, 0.25mM MgCl₂) for 1 h before the sections were incubated with primary antibodies at 4 °C overnight in blocking buffer. Sections were washed three times in PBST for 15 min each and then incubated with fluorochrome-conjugated secondary antibody at room temperature for 1 h. Sections were further washed three times in PBST and mounted using Prolong Gold anti-fade medium with DAPI (Life Technologies). Images were taken using a SPE (Leica) and processed using Imaris v4.8.

## Intracellular FACS staining

RFNG staining was performed on E11.5 AGM cell lysates. Single cell suspension was stained for the cell surface markers before the intracellular (rabbit anti-mouse) RFNG and a secondary antibody (donkey anti rabbit Alexa 488) was performed in FIX & PERM™ Cell Permeabilization Kit from Thermo Fisher (cat# GAS003). The samples were run on a Fortessa Instrument from BD Bioscience and analyzed with FlowJO V10. Statistical significance was determined with GraphPad prism 8.

## Proximity Ligation assay (PLA)

E10.5 and E11.5 CD1 or *Gfi1:tomato* trunks were cut into 150 μm thick sections, fixed in 2% Paraformaldehyde (Thermo Fisher) for 12 min, and permeabilized for 10 min in −20 °C 100% Methanol. Proximity Ligation assay was performed according to manufacturer's protocol (Sigma, cat# DUO96000). Briefly, after washing with PBS, the sections were blocked for 1 h with blocking solution and then incubated with the indicated primary antibody pairs as multiplex with N1/DLL4 (Sigma, Duolink DUO92008), N1ex/JAG1ex (sigma Duolink DUO92013), N1int/JAG1int (sigma, Duolink DUO92014) and (rat anti-mouse) CD31 or (rat anti-mouse) cKIT (please see antibody list in Supplementary Data 1). The thick sections were embedded in 80% glycerol in glass bottomed petri dishes (ibidi cat# 81156) and imaged with a SPE (Leica) instrument. Images were taken as z- stacks and rendered to a 3D representation using Imaris v4.8. PLA signals from cell images were manually counted (IAHC or 10 endothelial cells) and plotted using Prism (GraphPad prism 8).

## Nascent RNA capture assay

E11.5 *Gfi1:tomato* AGM cell suspension were stained for CD31 and cKIT. Endo (CD31+cKIT-GFI1-) or GFI1 + IAHC (CD31+cKIT+GFI1 + ) were FACS purified and treated with 10 µM of compound E (gamma-Secretase Inhibitor XXI, Merk cat # 565790) overnight. The following morning, the two samples were washed, separated as 300−400 cells/sample and incubated for 4 h in methylcellulose with the addition

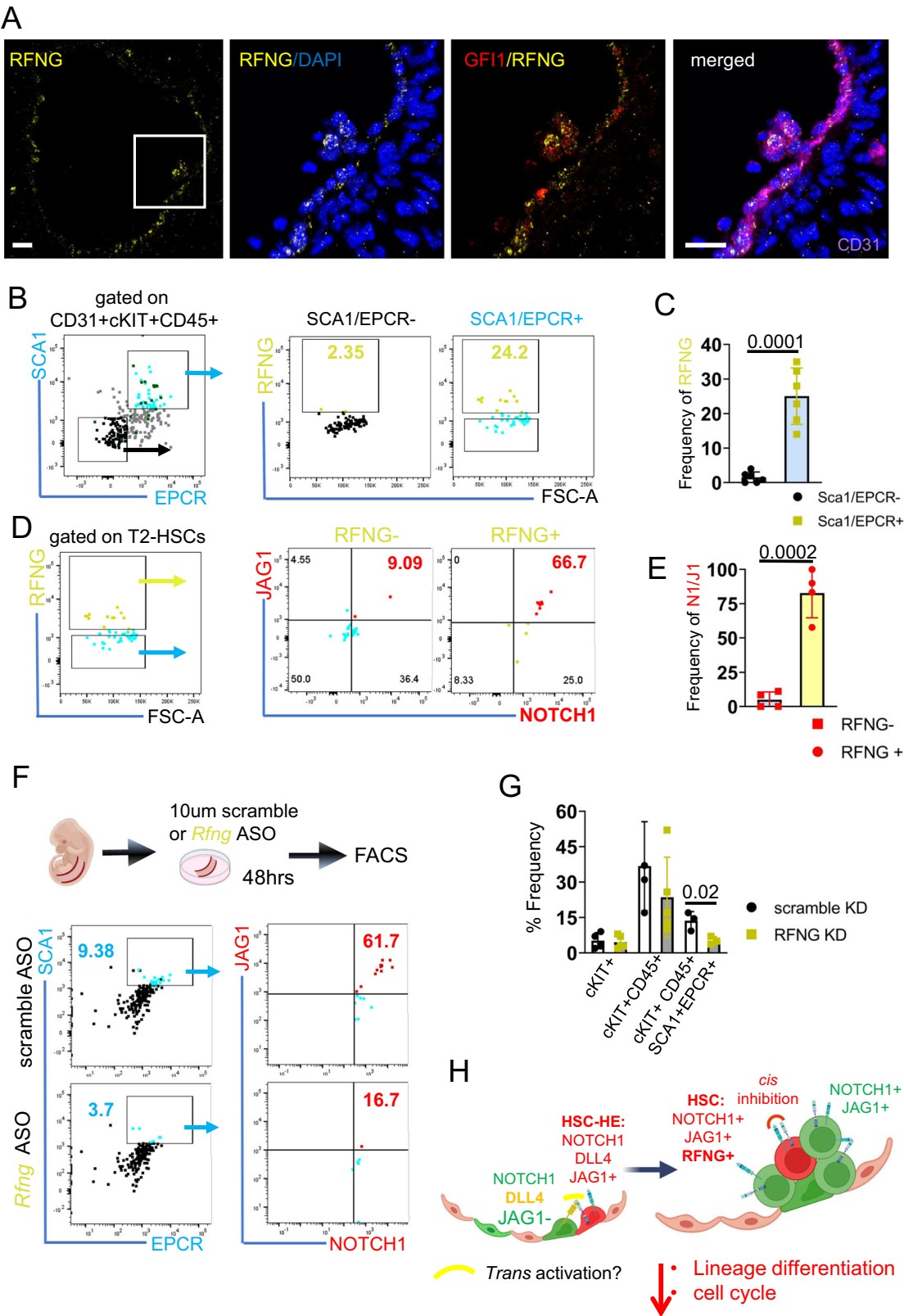

0.3 mM of EU (ethynyl Uridine, Thermo Fisher, cat #C10365) with 1ul DMSO, 10 μM of compound E or 4 μg/ml of recombinant Fc-JAG1 (R&D Systems, Cat #599-JG). After the EU labelling, cells were harvested and processed for purification of EU labeled RNA. Nascent RNA was purified from total nuclear RNA samples using the Click-iT Nascent RNA Capture Kit (Thermo Fisher, cat #C10365) according to the manufacturer's instructions.

**Nascent RNA capture- sample processing**

Nascent RNA was purified from total nuclear RNA samples using the Click-iT Nascent RNA Capture Kit (Thermo Fisher, cat #C10365) according to the manufacturer's instructions. In brief, biotin-azide was attached to ethylene-groups of the EU-labeled RNA using click-it chemistry and the pulled down of EU-RNA captured on the beads was immediately pre-amplified for 11 cycles (SuperScript™ IV Single

**Fig. 7 | NOTCH1/JAG1 cis interaction and HSC emergence is dependent on RFNG.**
**A** IHC on E11.5 AGM sections for RFNG (yellow), GFI1 (red), CD31 (magenta) and DAPI (blue). Representative image from 4 different IHC with $n = 4$ embryos. Scale bar = 10 μm. **B** Flow cytometry plots for Sca1 and EPCR after gating for CD31 + / cKIT + /CD45 + . SCA-/EPCR- or SCA + /EPCR+ and further analysed for RFNG. RFNG gate is superimposed onto Sca1 and EPCR plot. **C** Quantification of the frequency of RFNG in either SCA-/EPCR- or SCA + /EPCR+ fraction of CD31 + /cKIT + /CD45 + . Vertical error bars indicate the mean and standard deviation values. Statistical significance was calculated with two-tailed t-tests. (3 independent experiments with $n = 21$ embryos). Source data are provided as a source data file. **D** CD31 + /cKIT + / CD45+Sca1+EPCR + T2-HSCs were gated for RFNG expression. RFNG- and RFNG + T-HSCs were further plotted for JAG1 and NOTCH1. **E** Quantification of the frequency of JAG1/NOTCH1 in either RFNG- or RFNG+ fraction of T2-HSCs. Statistical significance was calculated with two-tailed *t*-tests with data derived from ($n = 3$ independent with $n = 21$ embryos). Vertical error bars indicate the mean and standard deviation values. Source data are provided as a source data file. **F** Cartoon of

experimental set up. E10.5 AGMs were cultured as explant with 10 μM control (scramble) FANA-ASO or 10 μM of RFNG FANA-ASO and processed for FACS analysis 48hrs later. Flow cytometry plots for Sca1 and EPCR after gating for CD31 + / cKIT + /CD45+ for scramble FANA-ASO or RFNG FANA-ASO. The T2-HSCs were then further analysed for JAG1 and NOTCH1 levels. **G** Quantification of the frequency of cKIT, cKIT+CD45+ and cKIT+CD45 + EPCR+Sca1 + T2-HSCs in either scramble or RFNG ASO. (n = 3 independent experiments, $n = 24$ embryos). Statistical significance was calculated with two-tailed t-tests. Vertical error bars indicate the mean and standard deviation values. Source data are provided as a source data file. **H** Scheme of proposed modulation of HSC specification through RFNG. GFI1 HE can be separated into HSC-HE that expresses higher levels of *Jag1*. Within the GFI1 + IAHC, some JAG1 positive cells co-express NOTCH1 on the surface, and this co-expression exists as a *cis* (inhibitory) conformation due to the presence of RFNG. RFNG expression is restricted to the T2-HSC population and this *cis* interaction is necessary to maintain stem cell identity by regulating cell cycle and differentiation-associated genes. (ASO: Antisense oligonucleotides).

---

Cell/Low Input cDNA PreAmp Kit, cThermo Fisher cat # 11752048) in accordance to manufacturer's instructions. Finally, double-stranded cDNA was purified using Agencourt AMPure XP beads (Beckman Coulter, cat #A63882). Validation PCRs were performed using the primers listed in Supplementary Data 1.

### Nascent RNASeq data analysis
Libraries were prepared at the Genomics Unit of PRBB (Barcelona, Spain) using Clontech SMARTer kit for low input material and cDNA was sequenced using Illumina NextSeq 2000 platform (50 bp single end reads). Samples sequencing depth ranged between 46 M and 53 M reads (average 49 M reads) per sample.

Quality control was performed on raw data with FASTQC tool (v0.11.9). Raw reads were trimmed to remove Clontech SMARTer IIA oligo (AAGCAGTGGTATCAACGCAGAGTAC) 5' presence with cuta-dapt (v4.2)[87]. Default parameters were used except for a maximum 5% error rate and no indels allowed. Trimmed reads were aligned to reference genome with STAR aligner tool (v2.7.8). STAR was executed with default parameters except for (i) the number of allowed mismatches was set to 1 and (ii) short reads consideration was relaxed to 50% of read length. Required genome index was built with corresponding GRCm38 gtf and fasta files retrieved from Ensembl (http://ftp.ensembl.org/pub/release-102/). Obtained BAM files with uniquely mapped reads were considered for further analysis. Raw gene expression was quantified using featureCounts tool from subRead software (v2.0.1) with gene as feature[88]. Obtained raw counts matrix was imported into R Statistical Software environment (v4.2.1) for downstream analysis. Raw expression matrix included 55,487 genes per 7 samples which were distributed in 3 different conditions: 2xFc-JAG1, 2xCompE and 3xWashed-out. Samples were considered as paired among conditions, referring them as three different biological samples. Prior to statistical analysis, those genes with less than 10 raw counts in at least two replicates from the same condition were removed. After pre-filtering, 10,311 genes were available for testing. For visualization purposes, counts were normalized by variance-stabilizing transformation method using local fit Type as implemented in DESeq2 R package (v1.38.3)[89]. To conduct PCA, normalized expression matrix was corrected per biological sample effect for corresponding with the function removeBatchEffect from limma R package (v3.54.2). Differential expression analysis (DEA) was conducted with DESeq2. Fitted statistical model included biological sample and condition as covariates. Pairwise comparisons for condition levels were tested considering the washed-out as the reference. DEGs were called with adjusted p-values (FDR) < 0.05 and expressed in at least two replicates in any of the two conditions being compared (minimum of 10 raw counts per sample).

### Functional analysis
Overrepresentation analysis was applied over lists of selected genes from scRNA-seq data or RNA-seq data analysis. The Gene Ontology (Biological Process ontology, GO BP terms) and KEGG PATHWAY databases for Mus Musculus were interrogated by means of cluster-Profiler R package (v4.6.2). Corresponding Entrez identifiers were used. Benjamini-Hochberg procedure was used to obtain adjusted p-values. Overrepresented GO BP terms (adjusted *p*-value < 0.05) were simplified using the simplify function from clusterProfiler with default parameters. A simplified list of terms was plotted on the semantic space obtained from REVIGO, executed with default parameters.

### Ex vivo culture of AGMs with scramble or RFNG FANA-ASO
AGMs of E10.5 embryos were dissected in PBS with 7% fetal calf serum (FCS) and penicillin/streptomycin (100 U/mL) and cultured as explants for 2 days in medium consisting of Stemspan (Stem Cell Technologies, cat # 09600), 20% fetal calf serum, L-glutamine (4 mM), penicillin/ streptomycin (50 units/ml), mercaptoethanol (0.1 mM), IL-3 (100 ng/ ml), SCF (100 ng/ml) and Flt3L (100 ng/ml). All growth factors were purchased from Peprotech. Tissues were maintained in 5% $CO_2$ at 37 °C. To knock down RFNG, we purchased custom designed ASO-FANA oligos targeting the mRNA of RFNG (Aum Biotech, USA). We used either 10 nM of control/scramble or RFNG FANA-ASO during the culture period.

### Quantification and statistical analysis
Statistical parameters, including number of events quantified, standard deviation, and statistical significance, are reported in the figures and in the figure legends. Statistical analysis has been performed using GraphPad Prism 8 software (GraphPad), and $P < 0.05$ is considered significant. Two-sided Student's t-test was used to compare differences between two groups.

### Reporting summary
Further information on research design is available in the Nature Portfolio Reporting Summary linked to this article.

## Data availability
Single cell RNA-Seq and nascent RNA-seq data: GEO accession number GSE230794. Source data are provided with this paper.

## Code availability
Custom codes used in the study are available via the GitHub repository (https://github.com/BigaSpinosaLab/HSC_cis_inhibition_Notch1_Jag1).

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

## Acknowledgements

We thank Sarah Bray and David Sprinzak for critical discussions about *cis* interactions. We thank Genentech for kindly gifting the anti-JAG1 and anti-NOTCH1 antibodies. We thank all members of Espinosa and Bigas laboratories for helpful discussions and technical support. All illustrations in the manuscript were created with BioRender.com. We also thank the animal facility, FACS facility, and genomic facility of the PRBB and CRUK Manchester Institute for their technical support. This work was funded by grants from SAF2016-75613-R and PID2019-104695RB-I00 from Agencia Estatal de Investigación (AEI) and SLT002/16/00299 from Department of Health, Generalitat de Cataluña and 2021 SGR 00039 from AGAUR, Generalitat de Catalunya. The work in the G.L. laboratory is supported by Blood Cancer UK (19014) and Cancer Research UK Manchester Institute Core Grant (C5759/A27412). The work in R.B. laboratory was supported by a grant from the Ministerio de Ciencia e Innovación (MCIN - PID2020-120252RB-I00). RT is a recipient of BP2016 (00021) and BP/MSCA 2018 (00034) fellowship programs from Generalitat de Catalunya/MSCA. E.P. is a recipient of RYC2019-026415-I. M.M. is a recipient of a grant from the Instituto Carlos III, grant number CA22/00011 (cofunded by the European Social Fund Plus, ESF+, and by the European Union).

## Author contributions

AB, LE, and RT conceptualized the study, designed the experiments, analyzed data and wrote the manuscript. RT, WHN, SG, DG, and FM performed experiments and analyzed data. MM, KI, YG, ZF, and XW analyzed the scRNAseq and nascent RNA-sequencing data. BG and FC supervised the scRNA seq analysis. ME, M-CF, EP, RB, and GL supervised data analysis. RT, AB, GL, and LE wrote the manuscript.

## Competing interests

The authors declare no competing interests.
