## [Peer Review File · Nature Communications]

Cis inhibition of NOTCH1 through JAGGED1 sustains embryonic Hematopoietic stem cell fateREVIEWER COMMENTS

Reviewer #1 (Remarks to the Author):

This manuscript investigates an important question, namely how Notch1 signaling contributes to the maturation of hemogenic endothelial hematopoietic stem-cell precursors (HE-HSCs) into hematopoietic stem cells (HSCs). The authors claim that Radical fringe (Rfng) potentiates Jag1 cis-inhibition of Notch1 and promotes the maturation of these cells into HSCs. It is evident that the authors have invested a lot of effort into the work that is presented, and there are several features of the model that are attractive. However, the data currently presented do not sufficiently support all of the claims made in the manuscript.

The investigators observe that cells co-expressing Notch1 and Jag1 characterize the HSC-HE compartment, and provide reasonable evidence that Notch1-Jag1 interactions occur in cis, whereas Notch1-Dll4 interactions occur in trans. The weaknesses lie in the data presented in Figures 5 and 6, where some of the experimental details are not clear, and where the noise in the data makes it still unclear whether the claim that Jag1 cis inhibition is guiding a cell fate decision is truly substantiated.

Major comments about the figures:

Figure 4. The authors use a proximity ligation assay to evaluate Notch1-DLL4 and Notch1-JAG1 interactions in AGM explants at day 10.5. They observe an increased number of Notch1-JAG1 foci in IAHC cells compared to Notch1-DLL4 foci. Analysis using probes for the extracellular domains gives a similar signal to the signal for the intracellular domains, supporting the idea that this interaction occurs in cis – that is, within the same cell. The authors also observe that treatment with JAG1-Fc reduces the signal for the intracellular PLA pair, consistent with their model. However, the authors also observe that non-specific IgG treatment also greatly reduces the PLA signal. Could the authors explain why treatment with IgG also interferes with the PLA signal for cis interaction?

Figure 5. The description of the experimental design for the studies in this figure are not clear enough. First, it is not clear whether the 4 hour Jag1-Fc treatment (panel A) was performed using Jag1-Fc immobilized on the TC plate, which would act as an agonist to induce production of Notch1 ICD (N1ICD), or if it was performed using soluble Jag1-Fc, which should act as an antagonist with little change in the signaling activity (because it would be expected to replace the endogenous, inhibitory Jag1 bound in cis). The data in panel B are very noisy, and it is not evident what effect the Jag1-Fc (in whatever form was used) actually has relative to the basal cis-inhibited state. Moreover, in the earlier figures, it appeared that the strongest reporter genes are Hey1/2, rather than Hes1 and GAGA1, which raises the question of why Hey1/2 were not analyzed in this experiment. Inspection of Panel F shows a very large batch effect for the Jag1-Fc treated samples – the genes highly expressed in the sample labeled Fc-JAG1_1 are expressed in low amounts in the Fc-JAG1_3 sample, and vice versa (the labels also raise the question of what happened to sample Fc_JAG1_2). These uncertainties make it difficult to conclude that the claims made based on these results are adequately justified.

One possible line of experimentation that might help would be to determine whether the Genentech anti-Notch1 NRR Ab mimics the cis-inhibitory effect being attributed to Jag1, as predicted by the model. One possible experimental approach might be to prevent cis-inhibition by providing a Jag1 ASO, determine whether that treatment affects the developmental trajectory, and if it does, determine whether treating with the Genentech anti-Notch1 NRR antibody restores the differentiation in the presence of that ASO, as would be predicted if Notch1 silencing is driving the cell fate decision.

Figure 6. Inspection of the data shows that very few cells were analyzed in the cell populations studied in D (which argues for the correlation of Rfng expression in Notch1 and Jag1 expressing cells) and in F (ASO treatment to try to infer causality). In addition, even the heterozygous presence of any of the three fringe genes appears to support normal hematopoietic development in mice (<https://doi.org/10.4049/jimmunol.1402421>), raising the question of why Rfng is a specific N-acetylglucosaminyltransferase candidate for potentiating this cis-inhibitory activity when it is not needed in the organism. In this reviewer's opinion, the effort to implicate Rfng detract from the study, rather than add to it, and I would recommend removing this part of the study and focusing instead on strengthening the mechanistic work compressed within Figure 5 instead. At a high level, which of the three fringes may (or may not be) involved is a minor detail, and the main insight worthy of publication would be if the cis-inhibitory role of Jag1 can be rigorously substantiated.

Reviewer #2 (Remarks to the Author):

In this manuscript, the authors present a series of experiments that support a novel mechanism for the regulation of Notch receptor signaling during HSC emergence from HE. Specifically, they provide multiple lines of evidence to show that cis interactions between Notch1 receptor and one of its canonical ligands, Jagged1, on the surface of newly emerging HSCs (which is augmented by R-Fringe expression in the same cells) promotes the emergence of phenotypic "Type 2" HSCs by limiting trans-activation of Notch1 receptor. Transcriptional analysis by scRNAseq further suggests that this process is mediated in part by regulation of downstream transcriptional programs involving cell cycle and hematopoietic differentiation. The findings are highly significant to the field in that they support a mechanism for Notch pathway regulation that appears to reconcile several published studies reporting contrasting roles of various Notch receptors, ligands, and modifiers during arterial EC/HE differentiation and the subsequent HE to hematopoietic transition required to give rise to the first HSC. While the novel studies provided are supportive of the proposed mechanism, and the data is generally robust, I have one primary concern below regarding the direct relevance to functional HSC in the absence of transplantation studies, as well as several minor concerns throughout the manuscript that should be addressed prior to the manuscript being suitable for publication. If these concerns can be adequately addressed, I believe the manuscript is of sufficient novelty and impact that would warrant publication in Nature Communications.

Major Concern: Although the manuscript describes effects on HSCs, these are characterized only by immunophenotype, and no functional (transplantation) assays are carried out. Although the markers used for FACS are consistent with those recently characterized to enrich for HSC activity at this stage in development, these populations are still not entirely pure, and thus may also contain a subset of HSC-independent MPPs, LMPs, and other progenitors in addition to long-term engrafting HSCs emerging in the IAHC of the AGM. In the absence of transplantation assays, the manuscript should address this limitation in the results/discussion section when interpreting the findings.

Minor concerns:

1) Page 4 (Intro): "Firstly, it can switch between different cell fates independent of trans activation through expressing cis ligands" Is this referring to cis-activation? If so, this should be stated more clearly.

2) Page 5 (Intro): "whereas JAG1 is detected in robustly in IAHC" – the first "in" should be removed.

3) Fig 1 A: GFI1 and Gfi1 are both used here, capitalization should be consistent through the figure and manuscript.

4) Fig 1B: This is described as a representative plot but is labelled as E10.25-E11.5. Does it represent a mix of embryos at different stages? It would also be helpful to show here or in supplement how GFI expression was gated to identify HE and GFI+ IAHC subsets.

5) Fig 1C: It would be helpful to clarify if what is shown is expression of each ligand (in the Y-axis?) of the total population gated as HE. Perhaps axis could be re-labelled with ligands instead of HE. The use of the green color here is confusing as green in A represents cells gated as HE. Does green here indicate cells that are positive for each ligand? If so, would be helpful to show isotype controls for these gates (FMO).

6) Fig 1C-D: How do the authors explain relatively low frequency of Notch1 receptor expression in HE, when Notch1 receptor activation is presumably required at this stage for HSPC fate? Is the increased Notch1 receptor expression frequency amongst IAHC compared with HE expected?

7) Sup Fig S2 is labelled with an (A) but there is no (B), etc. It is also not clear from the figure or legend what the top panel shows. Is this the total population before gating? Same comment as for Fig 1C, it would be helpful to label the axis for clarity (is y-axis each of the Notch ligands?). Again, isotypes should be shown, or at least show isotype cut-offs for gating strategy.

8) Page 6: It's not clear what supplemental Fig 3A has to do with Fig 1C and D.

9) Page 6: Figure S3A appears to have a much larger % of DLL4 in HE at E11.5 (~86%) than the range shown in Fig 1D. Please explain this discrepancy.

10) Page 6: There is reference to S2A-B, but there does not appear to be an S2B.

11) Fig S1. The relevance of Fig S1 is unclear (it appears that these are not gated based on GFI1 expression as in the main figures and remainder of supplemental figures, and this figure is not referred

to until after Fig S2 and S3). It makes it somewhat confusing to follow these figures in the context of the order and writing contained in the results section. Perhaps the order should be changed, more context should be provided to explain the importance of this figure and relevance to main figures, or this supplement should be removed altogether? The relatively higher expression of Jag1 in the EC population here (CD31+kit-) vs HE (GFI1+CD31+kit-) should also be discussed.

12) Page 7/Fig 2: “The top 25 marker genes from clusters 0, 1, 3 and 5 confirmed a high molecular similarity among them.” There does not appear to be a description of the presumed cell types represented by these clusters. How do they relate to the HE and HSC clusters labelled in UMAP space? Why do they represent a separate trajectory from the HSC branch appearing to arise from HE by pseudotemporal analysis?

13) Fig 2F-G: Why does it appear that J1 expression by scRNAseq is restricted to the “HSC-primed HE” population and low/absent in the HSC population, when the opposite is observed regarding surface J1 expression in Fig 1? Similarly, why does it appear by index analysis that relative Jag1 surface expression is decreased in HSC compared with HE? This seems to contradict the findings in Fig 1.

14) Fig 2G: The y-axis in the middle panel (DLL4) appears to be cut off. Also, what is meant by “FACS Index levels”? Is this the fluorescence intensity measured for each cell?

15) Fig 4C: Both sets of N1/DLL4 interactions (for EC and IAHC) are indicated by yellow dots. These should be represented by unique symbols for clarity.

16) Page 10/Fig S6D: For Tie2:MNFG AGM experiments, it would be helpful to quantify numbers of PLA foci compared with control AGM similar to what is shown in figure 4C. In figure S6D, it appears that PLA foci have extended into the subaortic mesenchyme in the Tie1:MNFG AGMs compared to control AGMs. The authors should explain the proposed etiology of this finding and if it is expected.

17) Figure 4 legend title seems incomplete: “Figure 4: NOTCH1 and JAG1 for a cis interaction in IAHC”

18) Figure 5/page 11-12: For experiments involving Jag1-Fc, how was the ligand presented (immobilized/attached to plates vs solubilized in media)? This does not appear to be described in the methods section. The method could affect interpretation of results as trans-activation is generally presumed to require an immobilized ligand (egs. surface/bead coating) whereas soluble ligand may be inhibitory to Notch receptor activation.

19) Fig 5B: An alternative explanation for these findings may be that the IAHC have already downregulated Notch receptors relative to EC and thus are not as susceptible to activation by Notch ligands. Did the authors attempt a similar experiment with DLL4 ligand or higher concentrations of ligands to see if the cis-inhibition could be overcome?

20) Fig 5F: It appears that there is significant discrepancy in relative expression of several genes shown between individual samples within each treatment group. With only two samples obtained per group, how did the authors validate that these genes were significantly differentially expressed based on

treatment, rather than the result of individual sample variation?

21) Fig 6B: FMO staining should be shown for each of the SCA1+EPCR+ and SCA1-EPCR- populations to demonstrate that the RFNG+ population does not represent an autofluorescent population or results from overlap in the fluorescent channels used for EPCR and SCA1.

22) Fig 6D: In the gating for RFNG amongst T2-HSC, there are an identical population of cells (in red) in both panels labelled as RFNG- and RFNG+. What are these cells meant to represent? Are they from the RFNG+ or RFNG- T2-HSCs? Why is the central panel labelled with a number (8.57) whereas the right is not labelled? What does this value represent?

23) Fig 6D: How do the authors reconcile the relatively low frequency of cells co-expressing N1/J1 (particularly in the RFNG- subset) in T2-HSC defined here based on EPCR and SCA1 expression compared with the data shown in Fig. 1 G-H that suggests most T2-HSC (defined by GF11 expression) co-express N1/J1 throughout E10-E11.5?

24) Page 14: "Altogether indicates that RFNG is important to ...". This sentence seems to be missing a phrase such as "these results"

25) Page 15: "Similarly, and in support for a opposing effects for DLL4 and JAG1 in HSC specification, AGMs treated with a DLL4 blocking antibody show enhanced HSC activity and reduN target gene expression." The 'a' between for and opposing should be removed. There also appears to be a typo in "reduN" – perhaps this should be "reduced Notch"

26) Is there a known hematopoietic phenotype of Rfng knockout embryos? Is there a specific defect in HSC activity or IAHC formation that would be predicted by the studies in this manuscript? If so, this information should be included in the discussion section.

REVIEWER COMMENTS

Reviewer #1 (Remarks to the Author):

This manuscript investigates an important question, namely how Notch1 signaling contributes to the maturation of hemogenic endothelial hematopoietic stem-cell precursors (HE-HSCs) into hematopoietic stem cells (HSCs). The authors claim that Radical fringe (Rfng) potentiates Jag1 cis-inhibition of Notch1 and promotes the maturation of these cells into HSCs. It is evident that the authors have invested a lot of effort into the work that is presented, and there are several features of the model that are attractive. However, the data currently presented do not sufficiently support all of the claims made in the manuscript.

The investigators observe that cells co-expressing Notch1 and Jag1 characterize the HSC-HE compartment, and provide reasonable evidence that Notch1-Jag1 interactions occur in cis, whereas Notch1-Dll4 interactions occur in trans. The weaknesses lie in the data presented in Figures 5 and 6, where some of the experimental details are not clear, and where the noise in the data makes it still unclear whether the claim that Jag1 cis inhibition is guiding a cell fate decision is truly substantiated.

Major comments about the figures:

Figure 4. The authors use a proximity ligation assay to evaluate Notch1-DLL4 and Notch1-JAG1 interactions in AGM explants at day 10.5. They observe an increased number of Notch1-JAG1 foci in IAHC cells compared to Notch1-DLL4 foci. Analysis using probes for the extracellular domains gives a similar signal to the signal for the intracellular domains, supporting the idea that this interaction occurs in cis – that is, within the same cell. The authors also observe that treatment with JAG1-Fc reduces the signal for the intracellular PLA pair, consistent with their model. However, the authors also observe that non-specific IgG treatment also greatly reduces the PLA signal. Could the authors explain why treatment with IgG also interferes with the PLA signal for cis interaction?

In this experiment (Figure 4G-I), we cultured AGM sections *ex vivo* for 4 hours, as opposed to the AGMs that were directly collected for PLA experiments (Figure 4 A-F). The *ex vivo* culture media contained cytokines (IL-3, SCF and Flt3L) and manipulation is likely affecting the basal levels of interactions that we observe. To avoid the comparison between both set ups, we clearly stated that in the text:

In these *in vitro* conditions, IgG treated explants showed reduced number of NOTCH1-int/JAG1-int interactions compared to untreated IAHC (compare Figure 4E and F with Fig 4H and I), however exposure to exogenous Fc-JAG1 further decreased the number of NOTCH1-int/JAG1-int interactions (Figure 4H and I).

Figure 5. The description of the experimental design for the studies in this figure are not clear enough. First, it is not clear whether the 4 hour Jag1-Fc treatment (panel A) was performed using Jag1-Fc immobilized on the TC plate, which would act as an agonist to induce production of Notch1 ICD (N1ICD), or if it was performed using soluble Jag1-Fc, which should act as an antagonist with little change in the signaling activity (because it would be expected to replace the endogenous, inhibitory Jag1 bound in cis).

During the 4-hour treatment in Fig 5 (now Fig 6), the recombinant Fc-JAG1 was added as a soluble factor, but within a semi-solid methylcellulose media providing a viscose/stiff culture condition, which could also affect the properties of Fc-JAG1.

Although Fc-JAG1 has been documented to have an inhibitory effect on Notch receptors (Sun et al, 2018; Small et al, 2001; Urs et al, 2008), our data indicates that Fc-JAG1 has a competitor effect on cis-JAG1 and is able to disrupt the interaction (Fig 4 G-I). To further understand the behaviour of Fc-JAG1 within the cluster cells, we have treated whole AGMs with Fc-JAG1 and then proceeded to FACS purify the CD31+cKIT+CD45+ expressing HSPC (enriched for HSCs) population for qPCRs for Notch signalling targets, demonstrating its ability to activate Notch in these conditions. We confirm that there is an activation effect on Hes1 and Gata2 after Fc-Jag1 exposure that is inhibited when Notch1 blockage is added. Moreover, we find a similar activation effect after anti-Jag1 incubation, suggesting that both Fc-Jag1 and anti-Jag1 are able to disrupt the cis-interaction. This is now included as a new Figure 5 and added in the text on page 11-12.

The data in panel B are very noisy, and it is not evident what effect the Jag1-Fc (in whatever form was used) actually has relative to the basal cis-inhibited state. Moreover, in the earlier figures, it appeared that the strongest reporter genes are Hey1/2, rather than Hes1 and GAGA1, which raises the question of why Hey1/2 were not analyzed in this experiment.

This is a challenging experiment for the number of cells that we get from the AGM and the quantity of nascent RNA collected. Cells are individualized and sparsely plated on semisolid media and altogether this is not the most efficient way to detect Notch activation, however it is the most direct way to detect effects due to disruption of cis-interactions. In addition, as shown in new Fig 5, we can demonstrate an activating effect of Fc-Jag1 (or anti-Jag1) on Notch-targets in AGM explants, supporting the disruption of cis-inhibition. Unfortunately, although we tested expression of Hey1 and Hey2, we did not detect sufficient levels in these conditions.

Moreover, the data in the heatmap (now Fig 6F) shows a general upregulation of Notch signalling genes after Fc-Jag1 incubation compared to CompE-washout, although some heterogeneity is observed between replicates, we do a general trend that is consistent within both independent replicates.

Inspection of Panel F shows a very large batch effect for the Jag1-Fc treated samples – the genes highly expressed in the sample labeled Fc-JAG1_1 are expressed in low amounts in the Fc-JAG1_3 sample, and vice versa (the labels also raise the question of what happened to sample Fc_JAG1_2). These uncertainties make it difficult to conclude that the claims made based on these results are adequately justified.

The reviewer is right. The data is indeed noisy for the reasons we mentioned above and we carefully tried to specifically focus on what is conserved between the samples and conditions in the different replicates. In fact, in the heatmap we are showing relative values between each sample with its wash out control. Although there is heterogeneity, heatmap shows a general up-regulation for FcJag1 samples compared to their corresponding washed-out ones.

As the reviewer spotted, we had 3 original samples, but the sequencing quality was too low for sample #2 and the information extracted was not reliable. We therefore did not include that sample pair in the analysis.

One possible line of experimentation that might help would be to determine whether the Genentech anti-Notch1 NRR Ab mimics the cis-inhibitory effect being attributed to Jag1, as predicted by the model. One possible experimental approach might be to prevent cis-inhibition by providing a Jag1 ASO, determine whether that treatment affects the developmental trajectory, and if it does, determine whether treating with the Genentech anti-Notch1 NRR antibody restores the differentiation in the presence of that ASO, as would be predicted if Notch1 silencing is driving the cell fate decision.

We thank the reviewers for this suggestion, although the experiment is tricky since it needs to be conducted *ex vivo*, and as the reviewer has already spotted, the NOTCH1-JAG1 cis interactions are reduced in these conditions. Still, we tested these suggestions with the two different approaches that are explained above, using Fc-JAG1 and anti-JAG1 and genetic deletion. This is now included in a new Figure 5 and in the text on page 11-12:

“To substantiate our hypothesis further, we undertook two additional approaches. In the first instance, we treated wild type AGMs as explants with Fc-JAG1, anti-JAG1, anti-NOTCH1, or combined Fc-JAG1 with anti-NOTCH1 for 4 hours and collected CD31+cKIT+CD45+ cells for notch targets gene expression profiling (Figure 5A, Suppl table T1). In agreement with our observations that the cis-interaction were reduced upon Fc-JAG1 treatment in PLA assays, we detected higher levels of Hes1 and Gata2 when treated with Fc-JAG1 or anti-JAG1 (Figure 5A). Next, we induced the genetic deletion of Jag1 in endothelial and hematopoietic cells by using a Ve-cadherin-CreERT2/Jag1floxed mouse line. We induced the deletion with 10uM of 4-OHT starting from E10.5 *ex vivo* and subsequently profiled the CD31+cKIT+CD45+ cells for notch targets gene expression (Figure 5Bi). As in the previous experiments, the levels of Hes1 and Gata2 were elevated in Jag1 deficient cells (Figure 5Bii). In both approaches, we also included anti-NOTCH1 with Fc-JAG1 or Jag1 deletion. In both instances, the higher levels of Hes1 and Gata2 in Fc-JAG1 or Jag1 deleted cells was reverted linking NOTCH1 as the receptor for the identified cis-interaction (Figure 5Aii and Bii).”

Figure 6. Inspection of the data shows that very few cells were analyzed in the cell populations studied in D (which argues for the correlation of Rfng expression in Notch1 and Jag1 expressing cells) and in F (ASO treatment to try to infer causality). In addition, even the heterozygous presence of any of the three fringe genes appears to support normal hematopoietic development in mice (<https://doi.org/10.4049/jimmunol.1402421>), raising the question of why Rfng is a specific N-acetylglucosaminyltransferase candidate for potentiating this cis-inhibitory activity when it is not needed in the organism.

This is an interesting point. In the study pointed out by the reviewer, only the lymphoid cell compartment was analysed and even in this study, the authors found that marginal zone B-cells could not be restored to WT levels by one allele of Rfng (“However, MZ B cell frequencies in spleen from mice expressing only one allele of *Mfng* or *Rfng* were significantly reduced, whereas mice expressing only *Lfng* were similar to *Fng* LMR mice”), arguing for cell lineage/ tissue specific dose requirement for Rfng.

Furthermore, the phenotype we describe here is very specific and only evident in a very rare population of cells, and specific defects on bona-fide HSC may only be obvious after serial transplantation. We are not aware of any study that has specifically analysed the HSC compartment in Rfng KO mice.

In this reviewer’s opinion, the effort to implicate Rfng detract from the study, rather than add to it, and I would recommend removing this part of the study and focusing instead on strengthening the mechanistic work compressed within Figure 5 instead. At a high level, which of the three fringes may

(or may not be) involved is a minor detail, and the main insight worthy of publication would be if the cis-inhibitory role of Jag1 can be rigorously substantiated.

We think that this additional, functional explanation advances our insight into the dynamics of HSC biology.

Reviewer #2 (Remarks to the Author):

In this manuscript, the authors present a series of experiments that support a novel mechanism for the regulation of Notch receptor signaling during HSC emergence from HE. Specifically, they provide multiple lines of evidence to show that cis interactions between Notch1 receptor and one of its canonical ligands, Jagged1, on the surface of newly emerging HSCs (which is augmented by R-Fringe expression in the same cells) promotes the emergence of phenotypic “Type 2” HSCs by limiting trans-activation of Notch1 receptor. Transcriptional analysis by scRNAseq further suggests that this process is mediated in part by regulation of downstream transcriptional programs involving cell cycle and hematopoietic differentiation. The findings are highly significant to the field in that they support a mechanism for Notch pathway regulation that appears to reconcile several published studies reporting contrasting roles of various Notch receptors, ligands, and modifiers during arterial EC/HE differentiation and the subsequent HE to hematopoietic transition required to give rise to the first HSC. While the novel studies provided are supportive of the proposed mechanism, and the data is generally robust, I have one primary concern below regarding the direct relevance to functional HSC in the absence of transplantation studies, as well as several minor concerns throughout the manuscript that should be addressed prior to the manuscript being suitable for publication. If these concerns can be adequately addressed, I believe the manuscript is of sufficient novelty and impact that would warrant publication in Nature Communications.

Major Concern: Although the manuscript describes effects on HSCs, these are characterized only by immunophenotype, and no functional (transplantation) assays are carried out. Although the markers used for FACS are consistent with those recently characterized to enrich for HSC activity at this stage in development, these populations are still not entirely pure, and thus may also contain a subset of HSC-independent MPPs, LMPs, and other progenitors in addition to long-term engrafting HSCs emerging in the IAHC of the AGM. In the absence of transplantation assays, the manuscript should address this limitation in the results/discussion section when interpreting the findings.

We have extended the paragraph on page 16 (in the Discussion section) to address this limitation of the study.

“We find enrichment of gene sets that include cell cycle, chromatin remodelling, lineage differentiation, and antigen processing/presenting genes that are upregulated in response to Fc-JAG1, strongly indicating that the cis confirmation of NOTCH1-JAG1 preserves a naïve stem cell fate phenotype, although transplantation assays are needed to functionally confirm the identity of these cells.”

Minor concerns:

1) Page 4 (Intro): “Firstly, it can switch between different cell fates independent of trans activation through expressing cis ligands” Is this referring to cis-activation? If so, this should be stated more clearly.

In this instance, we are referring to the general mechanism of expressing ligands in cis, regardless of cis-activation or cis-inhibition. We have now made this clear by extending the sentence to “through expressing ligands in *cis* that can be activating or inhibiting”.

2) Page 5 (Intro): “whereas JAG1 is detected in robustly in IAHC” – the first “in” should be removed.

We thank the reviewer for spotting this typo. We have now deleted the “in” in the manuscript.

3) Fig 1 A: GFI1 and Gfi1 are both used here, capitalization should be consistent through the figure and manuscript.

We thank the reviewer for this comment. We have now carefully revised the manuscript in this regard and have made the changes.

4) Fig 1B: This is described as a representative plot but is labelled as E10.25-E11.5. Does it represent a mix of embryos at different stages? It would also be helpful to show here or in supplement how GFI expression was gated to identify HE and GFI+ IAHC subsets.

We apologise for this confusion. The representative FACS plot shown in Figure 1B is indeed from E11.5 embryos. The label “E10.25-E11.5” is to indicate that AGM derived cells at different time point were collected for the single cell Index sorts and single cell RNA sequencing. We have now removed that label to minimize misunderstandings. In Suppl Figure S2A we have now also added the gating for GFI1 positive cells from CD31+cKIT- (HE) and CD31+cKIT+ (IAHC).

5) Fig 1C: It would be helpful to clarify if what is shown is expression of each ligand (in the Y-axis?) of the total population gated as HE. Perhaps axis could be re-labelled with ligands instead of HE. The use of the green color here is confusing as green in A represents cells gated as HE. Does green here indicate cells that are positive for each ligand? If so, would be helpful to show isotype controls for these gates (FMO).

The FMO controls for each gating are shown in Suppl Figure S2A (now Suppl Figure S2B) and also Suppl Figure S4. Indeed, the green colour for HE and then the NOTCH1, JAG1 and DLL4 positive cells within the HE population was chosen deliberately. We have now used different shades of green and magenta for the sub gating for NOTCH receptor and ligands from the main HE/GFI1+ IAHC population to clarify this further.

6) Fig 1C-D: How do the authors explain relatively low frequency of Notch1 receptor expression in HE, when Notch1 receptor activation is presumably required at this stage for HSPC fate? Is the increased Notch1 receptor expression frequency amongst IAHC compared with HE expected?

It is a very interesting point. Notch signalling seems to be at its highest levels in the HE population (Figure 3A) but is progressively dampened down. One possible explanation is that NOTCH1 on the surface of the HE population is more dynamic since it is cleaved and processed to NICD, whereas in the IAHC, where NOTCH1 and JAG1 are in cis, a state that is persistent, we can easily detect the NOTCH1 and JAG1.

7) Sup Fig S2 is labelled with an (A) but there is no (B), etc. It is also not clear from the figure or legend what the top panel shows. Is this the total population before gating? Same comment as for Fig 1C, it

would be helpful to label the axis for clarity (is y-axis each of the Notch ligands?). Again, isotypes should be shown, or at least show isotype cut-offs for gating strategy.

We apologise for this mistake. The top panel shows the FMO control for NOTCH1, JAG1 and DLL4 gated on the endothelial (CD31) population. We have not labelled the y-axis individually to have a more simplified and tidier figures since too many labels can distract from the focus of the panel, but we have now explained the gates better in the figure legend:

“GFI1 positive cells within CD31cKIT- (HE, green) and CD31+cKIT+ (GFI1+IAHC, purple) were sub gated for NOTCH1, DLL4 and JAG1 in 3132s (E10.25) AGM cell lysates. The HE and IAHC cells are superimposed onto the CD31/cKIT FACS plot.”

8) Page 6: It's not clear what supplemental Fig 3A has to do with Fig 1C and D.

They are representative FACS plots of the data plotted in Figure 1C and D. We have now included in the text. On page 6:

“On average, a large majority of cells in the HE subpopulation (80%) only express DLL4 on their surface, and about 15-20% are NOTCH1+ cells (Figure 1C and D; Suppl Figure S3A)”.

9) Page 6: Figure S3A appears to have a much larger % of DLL4 in HE at E11.5 (~86%) than the range shown in Fig 1D. Please explain this discrepancy.

This is true. To analyse the dynamic changes in NOTCH receptor and ligands from E10.25-E11.5, we used different Flow Cytometry approaches. Most of the analysis were performed with conventional FACS analyser (BD Fortessa) that allow simultaneous detection of 8-9 colours. Recently, our Flow facility also acquired a Cytex spectral Flow Cytometer that allows the detection of many more fluorochromes in one sample. We took advantage of this new instrument to probe the entire panel of our markers of interest, GFI1, HSC markers plus the notch receptors and ligands. The frequencies might be different since the spectral is more sensitive, but the overall trend, ie decrease in DLL4 positive cells still remains.

10) Page 6: There is reference to S2A-B, but there does not appear to be an S2B.

We apologise for not labelling the Suppl Figure S2 more clearly. We have now added the label.

11) Fig S1. The relevance of Fig S1 is unclear (it appears that these are not gated based on GFI1 expression as in the main figures and remainder of supplemental figures, and this figure is not referred to until after Fig S2 and S3). It makes it somewhat confusing to follow these figures in the context of the order and writing contained in the results section. Perhaps the order should be changed, more context should be provided to explain the importance of this figure and relevance to main figures, or this supplement should be removed altogether? The relatively higher expression of Jag1 in the EC population here (CD31+kit-) vs HE (GFI1+CD31+kit-) should also be discussed.

This figure is referred to in the first paragraph of the results section on page 6:

“We detected the presence of most of the ligands and receptors at E10.5 and a decrease at E11.5 was observed for NOTCH1 and DLL4 in the endothelium (CD31+cKIT-CD45-) and IAHC (CD31+cKIT+), as well as for JAG1 in CD45+IAHC (Suppl Figures S1and Suppl table T1). Based on the literature and our

previous work^{44,47,49,50}, we focused our subsequent analysis on NOTCH1, NOTCH2 receptors and JAG1 and DLL4 ligands.”

We started the study with an unbiased approach with assessing the distribution of Notch signalling elements within the endothelial and hematopoietic compartment of the AGM without the *Gfi1:tomato* line. We think it is valuable information, maybe not for our story, but for the general understanding of Notch signalling element distribution and abundance in different AGM sub-populations, ie CD31+cKIT-endo, CD31+cKIT+ IAHC and CD31+cKIT+CD45+ T2 enriches IAHC. This data can serve as reference in stances when the GFI1 reporter line is not available. Also, to our knowledge, this is first study to comprehensively track the dynamics of Notch signalling elements at the protein level in these cell populations.

12) Page 7/Fig 2: “The top 25 marker genes from clusters 0, 1, 3 and 5 confirmed a high molecular similarity among them.” There does not appear to be a description of the presumed cell types represented by these clusters. How do they relate to the HE and HSC clusters labelled in UMAP space? Why do they represent a separate trajectory from the HSC branch appearing to arise from HE by pseudotemporal analysis?

We and previous studies have not been able to fully characterise this population. Recent reports suggest that these cells might be more committed progenitors that are generated in the AGM at the same time as HSC. In fact, a few recent reports find evidence that HSC and progenitors arise independently in the AGM already ((eMPPs, Yokomizo et al, 2022; Patel et al, 2022). We therefore hypothesise that we have captured these cells in our FACS Index sort and RNA sequencing. In suppl table T2 we have compared each cluster against all the other cluster and KEGG for each of the cluster.

We can only hypothesise that the HSC-HE and the “progenitor” HE is more similar than the HSC and “progenitor” that take up opposing areas in the pseudo time. Cluster 1 in Figure 2 C-E, that connects the HSC-HE with the “progenitor” cluster (Cluster 0,1,3 and 5) mostly consists of cells sorted as GFI1 + HE (Figure 2B).

13) Fig 2F-G: Why does it appear that J1 expression by scRNAseq is restricted to the “HSC-primed HE” population and low/absent in the HSC population, when the opposite is observed regarding surface J1 expression in Fig 1?

This finding puzzled us initially as well. We were very surprised to see such specific expression of Jag1 that is mainly restricted to the HSC-HE yet we find JAG1 protein in IAHC by FACS (Figure 1), IHC (Suppl Figure S6A) and PLA assays (Figure 4). Our assumption is that JAG1 is highly upregulated in HSC-HE (maybe downstream of Notch activation) and it is downregulated after that. However, in IAHC, it is not endocytosed (because of Notch activation) but instead, it is in a stable *cis* complex with NOTCH1. Also, we cannot rule out that Jag1 is expressed in the HSC branch but at much lower levels than in the HSC-HE and it is not detected by scRNAseq.

Similarly, why does it appear by index analysis that relative Jag1 surface expression is decreased in HSC compared with HE? This seems to contradict the findings in Fig 1.

Although we enrich with GFI1 positive cells for HE and HSC containing IAHC populations, we are catching a lot of other cells as well. This is evident in Figure 2B, where cells that were sorted as GFI1+ HE and GFI1+IAHC are mostly resident in the main cluster, and not only in the HSC-HE and HSC cluster. Only through the Index sort combined with single cell RNA seq can we distinguish the HSC population from more committed cells. Thus, our findings by Index sort and Figure 1 are not directly comparable.

14) Fig 2G: The y-axis in the middle panel (DLL4) appears to be cut off. Also, what is meant by “FACS Index levels”? Is this the fluorescence intensity measured for each cell?

Yes, in this panel we have plotted the fluorescence intensity for each cell within the indicated population. We have now added a label (MFI, Mean fluorescence Intensity) for the y-axis to make this clearer.

15) Fig 4C: Both sets of N1/DLL4 interactions (for EC and IAHC) are indicated by yellow dots. These should be represented by unique symbols for clarity.

This is true. We have kept same colour for the N1-DLL4 signal but now have different symbols for EC and IAHC to differentiate between the two cell types.

16) Page 10/Fig S6D: For Tie2:MNFG AGM experiments, it would be helpful to quantify numbers of PLA foci compared with control AGM similar to what is shown in figure 4C. In figure S6D, it appears that PLA foci have extended into the subaortic mesenchyme in the Tie1:MNFG AGMs compared to control AGMs. The authors should explain the proposed etiology of this finding and if it is expected.

The Tie2:MNFG AGM experiment is a control experiment to show that our PLA assay work, ie we expected to see increased N1-DLL4 interaction in the Tie2-MFNG AGMs. As expected, we see an increase in N1-DLL4 interactions, or at least high intensity in the IAHC. We agree that quantification would be important, however all the embryo sections and images that we have for that experiment have a high intensity and it makes it impossible to reliably count. We hope that the reviewer appreciates the main message of this experiment as a PLA control.

About the foci extended into the subaortic mesenchyme, it could be due to Tie2 expression in pericytes (Teichert et al, 2017), cells that are being developed around the aorta that time, and it is possible that we see N1-DLL4 interaction there on the presence of MFNG. However, more work needs to be done to confirm that.

17) Figure 4 legend title seems incomplete: “Figure 4: NOTCH1 and JAG1 for a cis interaction in IAHC”

We thank the reviewer for spotting this typo. We have now changed the title to “NOTCH1 and JAG1 form cis interaction in IAHC”.

18) Figure 5/page 11-12: For experiments involving Jag1-Fc, how was the ligand presented (immobilized/attached to plates vs solubilized in media)? This does not appear to be described in the methods section. The method could affect interpretation of results as trans-activation is generally presumed to require an immobilized ligand (egs. surface/bead coating) whereas soluble ligand may be inhibitory to Notch receptor activation.

This is an important detail that has also been raised by reviewer 1. We added the Fc-JAG1 as a soluble ligand, but the cells/ligand were embedded in semi-solid/stiff matrix (methyl-cellulose) which may have result in a different effect than liquid media. We have now investigated the effect on Notch activation of soluble Fc-Jag1 added to AGM explants then proceeded to FACS purify the CD31+cKIT+CD45+ expressing HSPC population for qPCRs for Notch signalling targets.

We find that Fc-JAG1 in these conditions is able to activate Notch (likely by releasing cis-inhibition). We have now included this data in new Figure 5.

19) Fig 5B: An alternative explanation for these findings may be that the IAHC have already downregulated Notch receptors relative to EC and thus are not as susceptible to activation by Notch ligands. Did the authors attempt a similar experiment with DLL4 ligand or higher concentrations of ligands to see if the cis inhibition could be overcome?

Our work does not imply that Notch1 receptors cannot be downregulated or inactivated by other means, in fact, it has been clearly shown that Gpr183 directs NOTCH1 receptor degradation in after EHT (Zhang et al, 2015), while Sox17 likely induces Notch1 (Lizama et al, 2015). We have now included this possibility in the discussion, page 15:

“Moreover, Notch1 receptor levels are also tightly regulated during HE and EHT by Sox17 and Gpr183.”

However, our data demonstrates a new mechanism operating in the IAHC and likely HSC by which Notch1 ensures its inhibition, but also can provides a mechanism for IAHC/HSC to coordinate precise notch signaling activity levels within their surrounding (Nadagopalan et al,2019).

20) Fig 5F: It appears that there is significant discrepancy in relative expression of several genes shown between individual samples within each treatment group. With only two samples obtained per group, how did the authors validate that these genes were significantly differentially express based on treatment, rather than the result of individual sample variation?

We understand the concern. This is a challenging experiment for the number of cells that we get from the AGM and the quantity of nascent RNA collected. Cells are individualized and sparsely plated on semisolid media, altogether this is not the most efficient way to observe Notch activation, we believe it is the most accurate way to detect effects due to cis-interaction disruption. Moreover, we are only showing relative values between paired-sample from the same embryo pool. Although there is heterogeneity, heatmap shows a general up-regulation for FcJag1 samples compared to their corresponding washed-out ones.

To validate the nascent RNA experiment further, in terms of up-regulation of notch signalling targets, we have now tested the consequences of JAG1 loss in IAHC by genetic deletion of Jag1 (Ve-cad CreERT2/Jag1 Flox/flox) from E10.5 onward, FACS purified the CD31+cKIT+CD45+ expressing HSPC population for qPCRs for Notch signalling targets. This data is now included in new Figure 5.

21) Fig 6B: FMO staining should be shown for each of the SCA1+EPCR+ and SCA1-EPCR- populations to demonstrate that the RFNG+ population does not represent an autofluorescent population or results from overlap in the fluorescent channels used for EPCR and SCA1.

We understand the reviewer's concern. We have now included FMO controls for EPCR and SCA1 in Suppl Figure S7B.

22) Fig 6D: In the gating for RFNG amongst T2-HSC, there are an identical population of cells (in red) in both panels labelled as RFNG- and RFNG+. What are these cells meant to represent? Are they from the RFNG+ or RFNG- T2-HSCs? Why is the central panel labelled with a number (8.57) whereas the right is not labelled? What does this value represent?

We apologise for the mistake. This was an error caused by superimposing the N1-JAG1 gate on to the RFNG- and + plots. We have now revised the FACS plots.

23) Fig 6D: How do the authors reconcile the relatively low frequency of cells co-expressing N1/J1 (particularly in the RFNG- subset) in T2-HSC defined here based on EPCR and SCA1 expression compared with the data shown in Fig. 1 G-H that suggests most T2-HSC (defined by GF11 expression) co-express N1/J1 throughout E10-E11.5?

We have now revised the FACS plots in Figure 6D (now Figure 7D) that was incorrect in the initial manuscript. We apologise for the error. Also, we performed intracellular staining for RFNG after the samples were stained for the surface marker, including NOTCH1 and JAG1. In these instances, there is some expected loss of signal since the samples go through a fixation and permeabilization process. Thus, we would not expect the same level of staining efficiency as freshly stained samples.

24) Page 14: "Altogether indicates that RFNG is important to ...". This sentence seems to be missing a phrase such as "these results"

Thank you. We have now added "these results" in the manuscript.

25) Page 15: "Similarly, and in support for a opposing effects for DLL4 and JAG1 in HSC specification, AGMs treated with a DLL4 blocking antibody show enhanced HSC activity and reduN target gene expression." The 'a' between for and opposing should be removed. There also appears to be a typo in "reduN" – perhaps this should be "reduced Notch"

Thank you. We have now corrected this sentence for the typos.

26) Is there a known hematopoietic phenotype of *Rfng* knockout embryos? Is there a specific defect in HSC activity or IAHC formation that would be predicted by the studies in this manuscript? If so, this information should be included in the discussion section.

As reviewer 1 mentioned, RFNG levels control the lymphoid output in the thymus and spleen. Apart from this study, we are not aware of further studies investigating the role for RFNG in haematopoiesis, certainly not in IAHC and HSC biology. Most likely, no one has analysed specifically the HSC population in RFNG KO mice or the triple fringe KO (Song et al, 1016). Since the phenotype could be just manifested in a rare population of bona-fide HSCs, it could have been easily missed since it would only be observed after serial transplantations. But yes, it would be certainly interesting to study the HSC in RFNG KO mice. This is now included in the discussion (page 16) section:

“It is worth mentioning that deletion of all three *fng* genes has minor effects on adult hematopoiesis, however phenotypic and functional studies of *Rfng* KO embryonic HSC could provide critical insight into HSC specification in the AGM. It is not uncommon that HSC specific phenotypes are not detected unless studied in serial transplantation settings.”

References

- Lizama, C., Hawkins, J., Schmitt, C. et al. Repression of arterial genes in hemogenic endothelium is sufficient for haematopoietic fate acquisition. *Nat Commun* 6, 7739 (2015).
- Patel, S.H., Christodoulou, C., Weinreb, C., Yu, Q., da Rocha, E.L., Pepe-Mooney, B.J., Bowling, S., Li, L., Osorio, F.G., Daley, G.Q., et al. (2022). Lifelong multilineage contribution by embryonic-born blood progenitors. *Nature* 606, 747–753
- Nagarajan Nandagopal Leah A Santat Michael B Elowitz (2019) Cis-activation in the Notch signaling pathway *eLife* 8:e37880.
- Small, D. et al. Soluble Jagged 1 represses the function of its transmembrane form to induce the formation of the Src-dependent chord-like phenotype. *J. Biol. Chem.* 276, 32022–32030 (2001).
- Song Y, Kumar V, Wei HX, Qiu J, Stanley P. Lunatic, Manic, and Radical Fringe Each Promote T and B Cell Development. *J Immunol.* 2016 Jan 1;196(1):232-43.
- Sun, J., Luo, Z., Wang, G. et al. Notch ligand Jagged1 promotes mesenchymal stromal cell-based cartilage repair. *Exp Mol Med* 50, 1–10 (2018).
- Teichert, M., Milde, L., Holm, A. et al. Pericyte-expressed Tie2 controls angiogenesis and vessel maturation. *Nat Commun* 8, 16106 (2017).
- Urs, S. et al. Soluble forms of the Notch ligands Delta1 and Jagged1 promote in vivo tumorigenicity in NIH3T3 fibroblasts with distinct phenotypes. *Am. J. Pathol.* 173, 865–878 (2008).
- Yokomizo, T., Ideue, T., Morino-Koga, S., Tham, C.Y., Sato, T., Takeda, N., Kubota, Y., Kurokawa, M., Komatsu, N., Ogawa, M., et al. (2022). Independent origins of fetal liver haematopoietic stem and progenitor cells. *Nature* 609, 779–784.
- Zhang, P., He, Q., Chen, D. et al. G protein-coupled receptor 183 facilitates endothelial-to-hematopoietic transition via Notch1 inhibition. *Cell Res* 25, 1093–1107 (2015).

REVIEWERS' COMMENTS

Reviewer #1 (Remarks to the Author):

The authors have responded thoroughly to the comments of both reviewers. Nevertheless, there are limitations to the study because of the difficulty in collecting samples of sufficient sample quantity and quality, as the authors themselves acknowledge in their responses.

The authors are encouraged to add a “Limitations of the study” section at the end of the discussion that clearly spells out the key limitations noted by both reviewers. Most prominent among these concerns was the inherent noisiness in the nascent RNA experiment (Fig 5) because of the difficulty in acquiring samples with sufficient discriminating power, and the residual uncertainty about the role of Radical Fringe and why other studies had not identified a KO phenotype that might be suggested by a requirement of RFNG in HSC activity or IAHC formation. I would then be fully supportive of publication once this additional section is included.

Reviewer #2 (Remarks to the Author):

The authors have addressed most of my concerns. I understand the challenge of adding transplantation experiments to an already significant body of work and appreciate the addition of this caveat in the discussion.

I have one remaining comment regarding Fig 6F. Perhaps it would be better to show paired samples with relative expression for each pair, since there is such variability between replicate experiments. If the samples are paired, why are they labelled 1&2 for the control, and 1&3 for the JAG1 samples?

REVIEWERS' COMMENTS

Reviewer #1 (Remarks to the Author):

The authors have responded thoroughly to the comments of both reviewers. Nevertheless, there are limitations to the study because of the difficulty in collecting samples of sufficient sample quantity and quality, as the authors themselves acknowledge in their responses.

The authors are encouraged to add a “Limitations of the study” section at the end of the discussion that clearly spells out the key limitations noted by both reviewers. Most prominent among these concerns was the inherent noisiness in the nascent RNA experiment (Fig 5) because of the difficulty in acquiring samples with sufficient discriminating power, and the residual uncertainty about the role of Radical Fringe and why other studies had not identified a KO phenotype that might be suggested by a requirement of RFNG in HSC activity or IAHC formation. I would then be fully supportive of publication once this additional section is included.

We have now made changes in the manuscript to be more moderate regarding our claims of RFNG regulating the *cis* interactions between NOTCH1 and JAG1. These include the following changes:

- We have changed the title to “*Cis* inhibition of NOTCH1 through JAGGED1 sustains embryonic Hematopoietic stem cell fate”, thereby removing Rfng from the title
- We do not mention Rfng in the abstract
- Instead of demonstrating, we now use suggesting in this paragraph:
“RFNG knockdown AGMs have reduced numbers of HSCs and the NOTCH1-JAG1 *cis* interactions are reduced, ~~demonstrating~~ **suggesting** that RFNG favors NOTCH1-JAG1 in *cis* and that this conformation is essential for HSC maintenance.”
- We have changed from the following paragraph to dampen our claims:

“In agreement with our hypothesis, a decrease in RFNG in the AGM resulted in significantly fewer phenotypic HSCs (CD45+CKIT+SCA1+EPCR+) concomitant with a reduction in NOTCH1-JAG1 co-expression (Figure 7F-G, Supplementary Figure S7C). Altogether these results indicate that RFNG is important to preserve the NOTCH1-JAG1 *cis* interaction in some IAHC cells and maintain their HSC phenotype.”

to

“In agreement with our hypothesis, **a decrease in RFNG in the AGM was observed concurrently with significantly fewer phenotypic HSCs (CD45+CKIT+SCA1+EPCR+) and a reduction in NOTCH1-JAG1 co-expression (Figure 7F-G, Supplementary Figure 7C).** **Altogether these results indicate that RFNG might preserve the NOTCH1-JAG1 *cis* interaction in some IAHC cells and maintain their HSC phenotype.**”

- We have replaced the verb “facilitate” with “modulate” in the sentence in the discussion
...*cis* interaction between NOTCH1 and JAG1 that is especially relevant in T2-HSCs and ~~facilitated~~ **modulated** by RFNG that integrates and further explains the previous observations.

- Instead of “~~We demonstrate that the persistence of this *cis* NOTCH1-JAG1 is dependent on RFNG expression,” we now say “**We show that the persistence of this *cis* NOTCH1-JAG1 is linked to RFNG expression, a...**”~~
- Instead of “~~Accordingly, we find RFNG concentrated in sparse cells within a few IAHC by IHC and by FACS, we specifically detect them in EPCR+/Sca1+/ T2-HSC. Furthermore, inhibition of *Rfng* results in a reduction of NOTCH1-JAG1 co-expressing T2-HSCs.~~” we now say “**We find RFNG concentrated in sparse cells within a few IAHC by IHC and by FACS, we specifically detect them in EPCR+/Sca1+/ T2-HSC. Finally, AGMs after knock down of *Rfng* showed a reduction of NOTCH1-JAG1 co-expressing T2-HSCs.**”
- We have removed “~~Here, we have unraveled an important regulatory mechanism for Notch receptor and ligand interaction through *Rfng*.~~” from the final remarks in the discussion section.

We believe that we have now considerably toned down on our claims on the role of *Rfng* in the regulation of *cis* interactions between NOTCH1 and JAG1.

Reviewer #2 (Remarks to the Author):

The authors have addressed most of my concerns. I understand the challenge of adding transplantation experiments to an already significant body of work and appreciate the addition of this caveat in the discussion.

I have one remaining comment regarding Fig 6F. Perhaps it would be better to show paired samples with relative expression for each pair, since there is such variability between replicate experiments. If the samples are paired, why are they labelled 1&2 for the control, and 1&3 for the JAG1 samples?

There was indeed a JAG1 sample number 2, but it was discarded for quality reasons (extremely low mapping rate compared to the rest of samples).

The figure shows relative expression for each condition (2CompE and 2Fc-Jag1) compared to their washout counterpart (W1, W2 and W3, Supplementary Figure 6F) and are not relative to each other (CompE vs Fc-JAG1). We apologise for the confusion. In fact, 3 independent experiments were conducted, and conditions were analysed as paired samples. Thus, we have 2 independent CompE and 2 independent Fc-JAG1 experiments. We have now revised the figure to avoid confusion by removing the labelling (CompE 1_2 and Fc-JAG1 3) from Figure 6F. We have also revised the Figure legend to clarify the nature of the data.